# The *Chlamydia pneumoniae* effector SemD exploits its host's endocytic machinery by structural and functional mimicry

Fabienne Kocher [1], Violetta Applegate[2], Jens Reiners [2], Astrid Port[2], Dominik Spona[1], Sebastian Hänsch[3], Amin Mirzaiebadizi [4], Mohammad Reza Ahmadian[4], Sander H. J. Smits [2,5], Johannes H. Hegemann [1,6] ✉ & Katja Mölleken[1,6]

To enter epithelial cells, the obligate intracellular pathogen *Chlamydia pneumoniae* secretes early effector proteins, which bind to and modulate the host-cell's plasma membrane and recruit several pivotal endocytic host proteins. Here, we present the high-resolution structure of an entry-related chlamydial effector protein, SemD. Co-crystallisation of SemD with its host binding partners demonstrates that SemD co-opts the Cdc42 binding site to activate the actin cytoskeleton regulator N-WASP, making active, GTP-bound Cdc42 superfluous. While SemD binds N-WASP much more strongly than Cdc42 does, it does not bind the Cdc42 effector protein FMNL2, indicating effector protein specificity. Furthermore, by identifying flexible and structured domains, we show that SemD can simultaneously interact with the membrane, the endocytic protein SNX9, and N-WASP. Here, we show at the structural level how a single effector protein can hijack central components of the host's endocytic system for efficient internalization.

The obligate intracellular bacterial pathogen *Chlamydia pneumoniae* (*Cpn*) causes infections of the upper and lower respiratory tract[1,2]. A certain proportion of these can result in severe respiratory illnesses, such as pneumonia, asthma and chronic bronchitis, as well as multiple sclerosis, inflammatory arthritis, lung cancer and Alzheimer's disease[2–6].

*Cpn's* developmental cycle begins with the adhesion of the infectious elementary body (EB) to the host-cell's plasma membrane (PM), and its internalisation into a membrane-enclosed "inclusion". The initial, transient contact between EB and host cell enables chlamydial surface proteins, such as Pmp proteins and LipP, to stably bind and activate host-cell receptors that trigger receptor-mediated

internalisation[7–9]. However, engulfment of the EB requires a membrane vesicle that is three to four times larger in diameter than a classical endocytotic vesicle[10]. *Cpn* solves this problem by secreting several entry-related, early effector proteins directly into the host cell via its type-III-secretion system (T3SS). These include soluble factors, such as Cpn0572 (the homologue of *Chlamydia trachomatis (Ctr)* TarP), and proteins that bind to the host's PM, such as SemC and SemD[11–13]. By hijacking components of the host's endocytic machinery, early effectors trigger the formation of an intracellular membrane-enclosed vesicle that encompasses the EB[14–16]. The membrane-bound effectors SemC and SemD play a vital role in this process. Each possesses an amphipathic helix (APH) with high affinity for phosphatidylserine (PS),

[1]Heinrich Heine University Düsseldorf, Faculty of Mathematics and Natural Sciences, Institute for Functional Microbial Genomics, Düsseldorf, Germany. [2]Heinrich Heine University Düsseldorf, Faculty of Mathematics and Natural Sciences, Center for Structural Studies, Düsseldorf, Germany. [3]Heinrich Heine University Düsseldorf, Faculty of Mathematics and Natural Sciences, Center for Advanced Imaging, Düsseldorf, Germany. [4]Institute of Biochemistry and Molecular Biology II, Medical Faculty and University Hospital Düsseldorf, Heinrich Heine University Düsseldorf, Düsseldorf, Germany. [5]Heinrich Heine University Düsseldorf, Faculty of Mathematics and Natural Sciences, Institute of Biochemistry, Düsseldorf, Germany. [6]These authors jointly supervised this work: Johannes H. Hegemann, Katja Mölleken. ✉e-mail: johannes.hegemann@hhu.de

a specific phospholipid found in the inner leaflet of the PM[12,13]. The binding of SemC to PS induces extensive membrane curvature while SemD (382 aa) recruits and activates central endocytic host proteins[12,13].

Downstream of its N-terminal APH, SemD harbours two proline-rich domains (PRD1[91-100] and PRD2[117-122], Fig. 1a), the first of which binds to the SH3 domain of SNX9[12]. During classical endocytosis, SNX9, a BAR domain (bin-amphiphysin-rvs) protein, binds to the PM, induces membrane curvature and promotes vesicle closure[17–19]. Similarly, by recruiting SNX9 via SemD, *Cpn* amplifies membrane deformation at the site of EB entry and ensures the closure and maturation of the endocytic vesicle.

SemD also possesses two centrally located WH2 domains, which are involved in G-actin binding[12]. Furthermore, the C-terminal 165

residues (aa 218-382) of SemD are required for recruitment of N-WASP, an endocytic host protein that re-organises the actin cytoskeleton by interacting with the actin-branching complex Arp2/3[12]. N-WASP is a ubiquitously expressed member of the WASP family[20]. Signal reception and transduction of N-WASP are mediated by its basic region (BR), its GTPase-binding domain (GBD) and its verprolin-central-acidic (VCA) domain, respectively. The GBD domain consists of the Cdc42/Rac interactive domain (CRIB) and a C-sub motif (Fig. 2a)[21,22]. In resting cells, N-WASP resides in an autoinhibited cytosolic state mediated by intramolecular interactions between the GBD and VCA domains[22,23]. During endocytosis, Cdc42, a small GTPase belonging to the Rho family, is activated by guanine nucleotide exchange factors (GEFs) that catalyse the replacement of bound GDP by GTP[24]. Active, GTP-bound Cdc42 (Cdc42[GTP]) binds to the BR-GBD domain of N-WASP, and

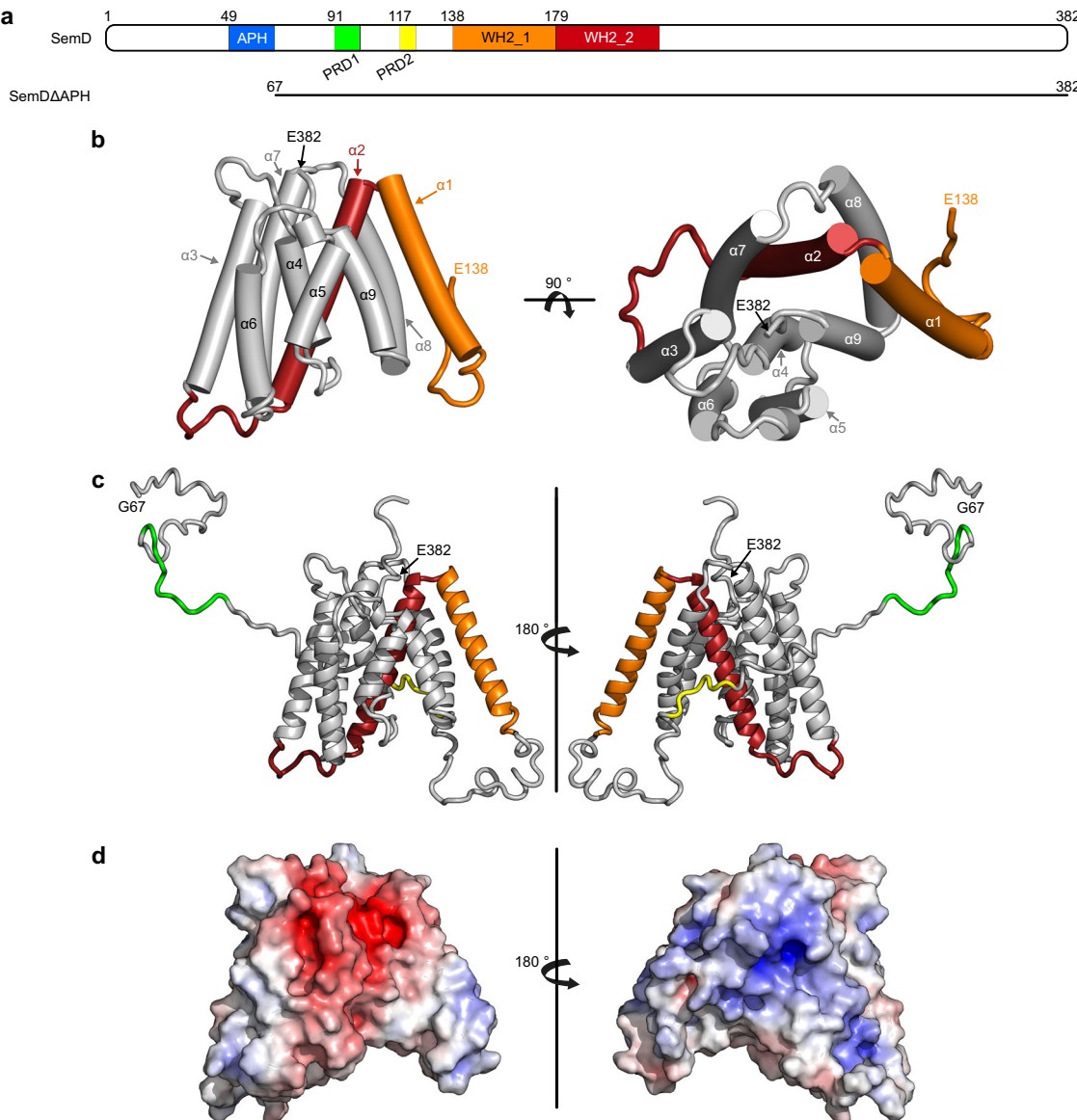

**Fig. 1 | Crystal structure of SemDΔAPH. a** Schematic representation of the primary structure of SemD, containing an APH[49-66], two proline-rich domains (PRD1[91-100], PRD2[117-122]) and two WH2 domains (WH2_1[138-178], WH2_2[179-216]). SemDΔAPH[67-382] is represented as a black bar. **b** The structure of SemDΔAPH as resolved by X-ray crystallography. The helices are depicted as cylinders and numbered from 1 to 9, starting at the N-terminus (α1- α9). In accordance with the colour code in **a**, the WH2_1 and WH2_2 are depicted in orange and red, respectively. The N-terminal E138 and the C-terminal E382 (both marked by black arrows) represent the first and last amino acids visible in the electron density. Right panel: 90° rotation. **c** SAXS best-fit CORAL model (χ² value of 1.197), based on the SemDΔAPH crystal structure, and including the added flexible tails (further models are shown in Supplementary Fig. 1h). PRD1 and PRD2 are coloured in green and yellow, respectively. Right panel: 180° rotation. G67 is the N-terminal amino acid, while E382, the last C-terminal residue of SemD, is followed by the C-terminal 10x-His-Tag. **d** Electrostatic surface representation of SemDΔAPH highlighting the negatively (red) and positively (blue) charged patches. Right panel: 180° rotation.

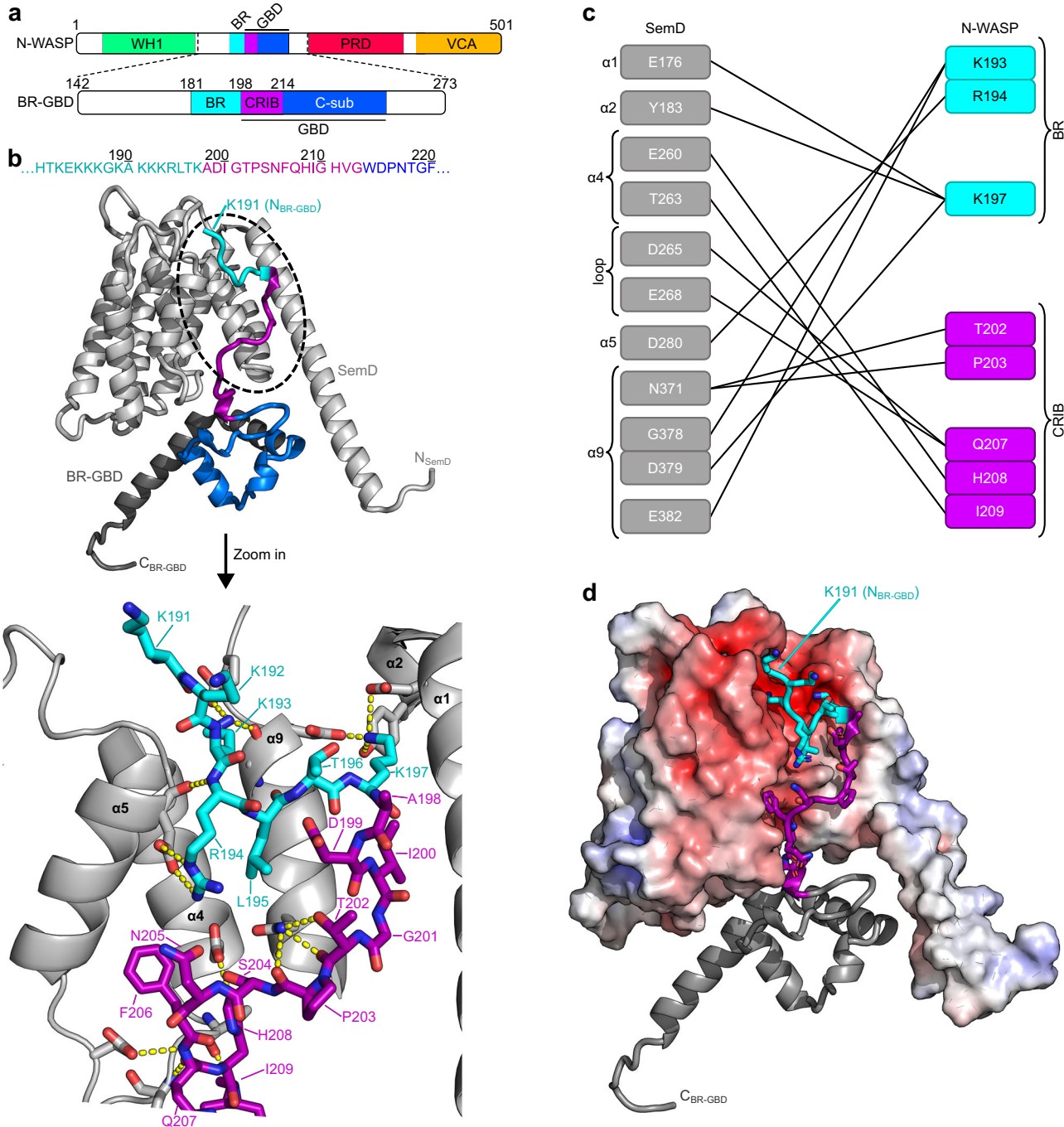

**Fig. 2 | SemD engages with BR-GBD in a Cdc42$_{GTP}$-mimicking manner.**
**a** Schematic representation of the N-WASP primary sequence. BR-GBD$_{142-273}$ was used for complex formation with SemDΔAPH. It contains the BR$_{181-197}$ domain (basic region, cyan), the CRIB$_{198-213}$ domain (Cdc42/Rac interactive binding motif, magenta) and the C-sub$_{214-250}$ domain (blue). **b** The structure of SemDΔAPH in complex with BR-GBD as resolved by X-ray crystallography, shown in cartoon representation. SemDΔAPH is shown in light grey, BR-GBD is coloured in dark grey with the BR domain in cyan, the CRIB domain in magenta and C-sub domain in blue.

The zoom in shows details of the binding of SemDΔAPH to BR-GBD. Important residues of SemDΔAPH and BR-GBD are shown in stick representation, while the rest of SemDΔAPH is shown as cartoon. Interactions (<3.5 Å) are shown by the yellow dashes. **c** Schematic representation of the detailed interactions between SemDΔAPH and BR-GBD. **d** Electrostatic representation of SemDΔAPH, highlighting the negatively charged patch in red and positively charged surface areas in blue. BR-GBD is coloured in dark grey (cartoon) with the BR domain in cyan and the CRIB domain in magenta, both depicted with stick residues.

triggers the release of the VCA domain, which in turn binds and actives the Arp2/3 complex (Supplementary Fig. 2b)[23,25]. Moreover, the BR binds to PI(4,5)P$_2$ in the inner leaflet of the PM, and recruits the actin polymerisation machinery to the site of endocytosis[26].

Cdc42 plays a central role in a large number of diverse biological processes such as the cell cycle, controlling gene transcription, regulating the cytoskeleton, cell movement and polarisation, hence

being a target for many virulence factors secreted by bacterial pathogens[27–29]. These factors modulate the activity of Cdc42 by mimicking host regulators such as GEFs, GTPase activating proteins (GAPs) and guanine dissociation inhibitors (GDIs), or by covalently modifying Cdc42[30–34]. In addition, bacterial effector proteins can bind the autoinhibited Cdc42-binding domain of N-WASP, thereby initiating actin polymerisation[35].

During a *Cpn* infection, the C-terminus of the membrane-bound SemD interacts with the BR-GBD domain of N-WASP, thus triggering N-WASP activation and Arp2/3-mediated actin polymerisation via an unknown mechanism[12]. This ensures the provision of branched F-actin bundles required for extensive membrane deformation and maturation of the EB-containing vesicle.

In this work, we elucidate the mechanism involved by determining the three-dimensional structure of SemD, alone and in complex with its host interaction partners. We demonstrate that SemD, a protein of 382 aa, can interact simultaneously with the PM, SNX9 and N-WASP, thereby combining membrane association and deformation with modulation of the actin polymerisation machinery. Using small-angle X-ray scattering (SAXS), crystallography and mutational analysis, we show that SemD structurally and functionally mimics the activation of N-WASP by $Cdc42_{GTP}$, thus enabling *Cpn* to activate N-WASP in a $Cdc42_{GTP}$-independent manner. Further, by using pulldown assays and stopped-flow experiments, we show that SemD binds N-WASP more tightly than $Cdc42_{GTP}$, and that SemD is a specific N-WASP activator, not binding to formin like-protein L2 (FMNL2), another $Cdc42_{GTP}$-target protein. Our structural data also reveal that the N-WASP binding region of SemD is separated from its PRD1 – which is responsible for SNX9-SH3 binding – and from the membrane-binding APH domain via flexible linker regions. These features permit highly adaptable rearrangements of the individual binding sites, which reduce steric hindrance and facilitate simultaneous binding of the PM, SNX9 and N-WASP. These concurrent interactions enable *Cpn* to rapidly modulate the PM and the actin cytoskeleton, which ensures the successful formation of a large endocytic vesicle, and the rapid uptake of the EB within 15 minutes after its initial adhesion to a non-phagocytic host cell.

## Results

### SemD folds into a multipurpose interaction structure

To elucidate how SemD functions at the molecular level, we determined the 3D structure of the protein, N-terminally truncated up to and including the APH motif ($SemD\Delta APH_{67-382}$) at a resolution of 2.1 Å (Fig. 1a, b, Supplementary Fig. 1a, b and Supplementary Table 1). The resulting structure revealed that the C-terminal portion of $SemD\Delta APH$ (aa 138-382) folds into a rigid core, consisting of nine α-helices, which is N-terminally flanked by a long intrinsically disordered region (IDR, aa 67-137, Fig. 1b). Owing to its flexibility, the latter is not visible in the electron density. The proline-rich domains (PRD1 and PRD2) are within the IDR and provide a highly flexible interaction surface. The first and second α-helices harbour the WH2_1 and WH2_2 domains involved in G-actin binding[12] (Fig. 1b). Within the electron density, the amino acids are clearly visible, except for the connecting loop between helices 2 and 3, which is presumably flexible; here, the side-chains were not included in the final model. Although structural comparisons using EBI-fold revealed similar proteins (all with a root-mean-square deviation (RMSD) > 3.5 Å), no informative conclusions could be reached, since the only feature shared between them was a high helical content.

To clarify how $SemD\Delta APH$ behaves in solution, we performed SAXS analysis. We found that $SemD\Delta APH$ is a monomer in solution (Supplementary Table 2) and the *p(r)* function indicated a globular core – corresponding to the helical core domain revealed by the X-ray structure – and an elongated tail (Supplementary Fig. 1c–g). We calculated the theoretical scattering of the solved $SemD\Delta APH$ crystal structure and compared it with the experimental data for $SemD\Delta APH$ in solution. The resulting CRYSOL fit yielded a $\chi^2$ value of 14.63 (an indicator on how well the model fits to the scattering curve in solution) and showed a high mismatch in the low s region (Supplementary Fig. 1c). This is not surprising, because the N-terminal residues (aa 67-137) are not solved in the crystal structure and the *p(r)* function showed an elongated tail, most probably the N-terminal region. Based on the information derived from SAXS data, we added the missing N-terminal

residues (aa 67-137) to complete the $SemD\Delta APH$ structure (Fig. 1c). The resulting models showed that these N-terminal residues comprise the IDR tail (the best-fit model is shown in Fig. 1c, an overlay of independent models, showing the same tendency of the tail orientation, are shown in Supplementary Fig. 1h). This resulted in an improved $\chi^2$ value of 1.20 (Supplementary Table 2). Furthermore, electrostatic analysis of the rigid core of SemD revealed a large, negatively charged patch on the front of the protein and a smaller positively charged patch on the back (Fig. 1d).

Taken together, the combined crystallographic and SAXS-based structure of $SemD\Delta APH$ reveals that its N-terminal segment, with which SH3 domain-containing proteins interact, is flexible. The nine α-helices that constitute the rigid core include the two WH2 domains, involved in G-actin binding, and the N-WASP interaction site[12]. Electrostatic analysis of the rigid core reveals two highly charged patches; a negative patch on the front and a positive patch on the back.

### SemD structurally mimics $Cdc42_{GTP}$ for N-WASP activation

Generally, $Cdc42_{GTP}$ activates N-WASP by binding to its BR-GBD domain, which leads to the release of the VCA domain of N-WASP. The VCA domain then recruits the Arp2/3 complex, initiating actin branching and polymerisation (Supplementary Fig. 2b). However, during *Cpn* uptake, secreted SemD, whose APH domain interacts with the PM, binds to N-WASP and activates it in an as yet unknown fashion, thus initiating the formation of branched F-actin structures upon recruitment of the Arp2/3 complex[12]. Intriguingly, it has been shown that, when bound to synthetic membranes via its APH domain, SemD is capable of binding and activating N-WASP[12]. The interaction between the two proteins requires the C-terminal part of SemD (aa 218-382) and the BR-GBD segment of N-WASP (Fig. 2a)[12]. To understand the activation of N-WASP by SemD, we structurally analysed the complex formed by recombinant $SemD\Delta APH$ and the BR-GBD domain of N-WASP. To this end, we purified the recombinant proteins separately, allowed them to interact and isolated the resulting complex by size exclusion chromatography (SEC, Supplementary Fig. 2a). The elution fractions containing the complex were pooled and analysed by both crystallography and SAXS (Fig. 2b, d).

The co-crystal of $SemD\Delta APH$ in complex with BR-GBD yielded a structure with a medium resolution of 3.3 Å, which is attributable to the flexible termini of both proteins. Interestingly, the loop connecting helices 2 and 3 ($_{208}$GTSSTG$_{-213}$) of $SemD\Delta APH$ is stabilised in the complex and can now be modelled. Despite the moderate resolution of the crystals containing the $SemD\Delta APH$ – BR-GBD complex, the interaction surface between the two proteins is well resolved. Comparison of the structures of $SemD\Delta APH$ alone and in complex with BR-GBD revealed virtually identical conformations of the SemD core regions, as indicated by a low RMSD of 0.7 Å. Next, we identified intermolecular contact sites between $SemD\Delta APH$ and BR-GBD by identifying the amino acids that were closer than 3.5 Å to each other[36]. We found three positively charged residues within the BR domain of N-WASP (K193, R194 and K197) that interact directly with the negatively charged area found on the front of SemD, formed by helices α1, α2, α5 and α9 (Fig. 2b–d). Additionally, five residues of the N-WASP CRIB domain engage with residues in the SemD binding groove, mainly formed by helix α4, the adjacent loop and helix α9 (Fig. 2b, c). The N-WASP C-sub domain is located underneath the helical arrangement of $SemD\Delta APH$, flanked by its extended helix α1 (Fig. 2b, d). Owing to the flexibility of the N-terminal domains of $SemD\Delta APH$ and BR-GBD, these regions are not resolved in the crystal structure of the complex.

Using SAXS, we validated the results obtained by crystallography. Analysis of BR-GBD alone showed that, in solution, it forms a monomer with a globular central region, flanked by a highly flexible N-terminal domain (Supplementary Fig. 3a–d, Supplementary Table 2). SAXS analysis of the $SemD\Delta APH$ – BR-GBD complex confirmed 1:1 stoichiometry in solution (Supplementary Table 2), in accordance with the

crystal structure. Analysis of the *p(r)* function and the dimensionless Kratky plot showed, that, upon formation of the complex, the unstructured segment of the BR-GBD does not adopt a specific secondary structure, but must take on a more constrained posture to bind to the core region of SemDΔAPH (Supplementary Fig. 3e–j), while the other domains retain their flexible characteristics. To confirm the position of the globular C-sub domain of N-WASP, we used the ensemble optimization method (EOM). Here, the interaction surface between SemDΔAPH and the N-WASP BR and CRIB segments found in the crystal structure is fixed, and the C-sub domain is allowed to vary freely in 3D space. Under these conditions, the conformation of the C-sub domain is the same as that seen in the crystal structure. Moreover, the final EOM models of the complex formed by SemDΔAPH and BR-GBD revealed, that an elongated complex conformation is preferred, which is attributable to the flexible N-termini of SemDΔAPH and BR-GBD (Supplementary Fig. 3k).

Taken together, our structural data show that the interaction of SemD with N-WASP requires the CRIB domain, together with the five C-terminal BR residues (aa 193-197) three of which are positively charged. Three of the five amino acids interact with the negatively charged patch on SemD, while the CRIB domain is embedded in the binding groove provided by SemD (Fig. 2).

## SemD binds the N-WASP BR-GBD in a bipartite fashion

As described above, SemD binds to N-WASP by interacting with positively charged residues of the BR and with the CRIB domain. Moreover, structurally speaking, this mechanism is very similar to the interaction of WASP with Cdc42$_{GTP}$[37]. WASP and N-WASP belong to the same protein family, share 56% identity and 74% similarity, and have strikingly similar domain architectures and regulatory mechanisms[23]; however, the structure of the N-WASP – Cdc42$_{GTP}$ complex is not available thus far. Both proteins are crucial for transducing cell surface signals to the actin cytoskeleton, but while N-WASP is expressed ubiquitously, WASP is only present in non-erythroid hematopoietic cells[38]. To activate WASP (and N-WASP), Cdc42$_{GTP}$ interacts with the C-terminal residues of the BR domain – starting at the KKK$_{230-232}$ motif (corresponding to KKR$_{192-194}$ in N-WASP) – and with the CRIB domain (Fig. 3a)[23]. Mutational analysis indicated that the binding is strongly impeded when the KKK$_{230-232}$ motif in WASP is replaced by either uncharged or negatively charged amino acids[23,39]. Interestingly, with the C-terminal BR residues engaged in Cdc42$_{GTP}$ binding, the N-terminal BR residues can simultaneously bind to membranes containing PI(4,5)P$_2$, its preferred lipid[23].

Given the high structural similarity between the modes of interaction used by SemD and Cdc42$_{GTP}$ to bind and activate N-WASP, we tested for functional mimicry. To do so, we constructed deletion variants of the N-WASP BR-GBD domain lacking either the BR (BR-GBDΔ) or both the BR and the CRIB domain (BR-GBDΔΔ) (Fig. 3b). We also investigated whether or not the N-terminal BR residues mediate binding to PI(4,5)P$_2$ when SemD occupies the C-terminal BR residues. As an experimental setup, we chose to use giant unilamellar vesicles (GUVs) as a synthetic membrane model of the PM.

To test for recruitment of the individual BR-GBD variants by SemD, we used PS-containing GUVs, which mimic the lipid preferentially bound by SemD. For quantification, we calculated the fluorescence intensity ratio by measuring the maximal intensity at the perimeter of the GUV and setting it in relation to the average background intensity outside the GUV. In our control experiments, GFP alone showed no unspecific binding to GUV-bound SemD labelled with rhodamine (SemD$^{Rhod}$), and BR-GBD fused to GFP (BR-GBD$_{GFP}$) showed no binding to GUVs (Fig. 3c). Upon incubating BR-GBD$_{GFP}$ with SemD$^{Rhod}$ and GUVs, immediate colocalization of both proteins at the perimeter of the GUV was observed, indicating rapid and direct binding of BR-GBD$_{GFP}$ to GUV-bound SemD$^{Rhod}$ (Fig. 3c). Quantification revealed that binding persisted over a period of

20 min, and that fluorescence intensity at the GUV perimeter is $44.5 \pm 15.3$-fold higher than the average background fluorescence outside the GUV (Fig. 3d). Interestingly, upon incubation of BR-GBDΔ$_{GFP}$ (lacking the BR domain) with SemD$^{Rhod}$ and GUVs, weak binding was visible after 5 min, which significantly increased over the next 15 min. However, even after 20 min, the fluorescence intensity ratio was more than 18-fold lower than the signal obtained for BR-GBD$_{GFP}$ (Fig. 3c, d). Finally, we also examined the binding of BR-GBDΔΔ$_{GFP}$ (lacking both BR and CRIB) to SemD$^{Rhod}$. Quantification revealed a low fluorescence intensity ratio of $1.1 \pm 0.2$, which did not change over the next 20 min. Comparison of the data for BR-GBDΔΔ$_{GFP}$ with the negative control GFP revealed no significant difference in fluorescence intensity ratio, at $1.1 \pm 0.4$. We therefore concluded that BR-GBDΔΔ, which lacks both BR and the CRIB domain, shows no recruitment to GUV-bound SemD$^{Rhod}$.

Next, we tested for simultaneous binding of BR-GBD$_{GFP}$ to PI(4,5) P$_2$ and SemD$^{Rhod}$ by using PI(4,5)P$_2$-containing GUVs. Indeed, BR-GBD$_{GFP}$ alone bound to PI(4,5)P$_2$-containing GUVs while SemD$^{Rhod}$ alone showed only a very weak colocalization to PI(4,5)P$_2$-containing GUVs (Supplementary Fig. 2c). When SemD$^{Rhod}$ was incubated with PI(4,5)P$_2$-bound BR-GBD$_{GFP}$, immediate colocalization of both proteins was observed at the perimeter of the GUV, suggesting that BR-GBD can indeed interact with both, lipids and SemD, simultaneously, in a manner similar to that of the N-WASP BR segment upon its interaction with Cdc42$_{GTP}$.

Taken together, these data imply that SemD not only structurally but also functionally mimics Cdc42$_{GTP}$ to recruit, bind and activate the central endocytic host protein N-WASP. Thereby, SemD binds to the BR-GBD via a bipartite interaction, which involves (i) the binding of positively charged amino acids located in the C-terminal BR domain to the negatively charged patch on SemD and (ii) the insertion of the CRIB domain into the binding groove provided by SemD. Thus, during *Cpn* uptake, the secreted and PM-bound SemD recruits N-WASP and abrogates its intramolecular autoinhibition by mimicking the Cdc42$_{GTP}$ activity in structure and function, leading to VCA release and finally to Arp2/3-mediated F-actin branching.

## SemDΔAPH outcompetes Cdc42$_{GppNHp}$ for binding to N-WASP

So far, we have shown that SemD structurally and functionally mimics Cdc42$_{GTP}$ to bind and activate N-WASP. During a *Cpn* infection, PM-bound SemD redirects N-WASP function to the bacterial entry site thus mimicking Cdc42$_{GTP}$ for N-WASP binding. To compare N-WASP binding to SemD and Cdc42$_{GTP}$, respectively, we performed in vitro GFP-Trap® pulldown assays, using BR-GBD$_{GFP-His}$ as bait. BR-GBD$_{GFP-His}$ was incubated with SemDΔAPH$_{His}$, or with Cdc42 bound to a non-hydrolysable GTP analogue (Cdc42$_{GppNHp}$)[23,40], or with equimolar amounts of SemDΔAPH$_{His}$ and Cdc42$_{GppNHp}$. Following pulldown, we probed the composition of the flow through (FT) and elution fractions using immunodetection (Fig. 4a) and assessed the binding efficiency to BR-GBD$_{GFP-His}$, by correlating the band intensity of the elution to that of the FT (Elution:FT ratio, Fig. 4b). The negative control GFP indicated no unspecific binding to neither SemDΔAPH$_{His}$ nor Cdc42$_{GppNHp}$ (Supplementary Fig. 4b). Conversely, the positive controls indicated evident binding of SemDΔAPH$_{His}$ and Cdc42$_{GppNHp}$ respectively, to BR-GBD$_{GFP-His}$, with a quantified Elution:FT ratio of ~ 100% for each (Fig. 4a, b). Moreover, when equimolar ratios of SemDΔAPH$_{His}$ and Cdc42$_{GppNHp}$ were added simultaneously to BR-GBD$_{GFP-His}$, SemDΔAPH$_{His}$ showed an Elution:FT ratio of ~ 100 %, comparable to the positive control, while Cdc42$_{GppNHp}$ was now detected only in the FT (Fig. 4a, b). This experiment indicates that BR-GBD preferentially binds SemDΔAPH in the presence of active Cdc42. This result was confirmed and extended by stopped-flow measurements, in which addition of SemDΔAPH$_{His}$ to a preformed BR-GBD$_{His}$ – Cdc42$_{GppNHp}$ complex led to dissociation of the latter (Fig. 4d), while Cdc42$_{GppNHp}$ in the absence of SemDΔAPH binds to BR-GBD$_{His}$ on a millisecond

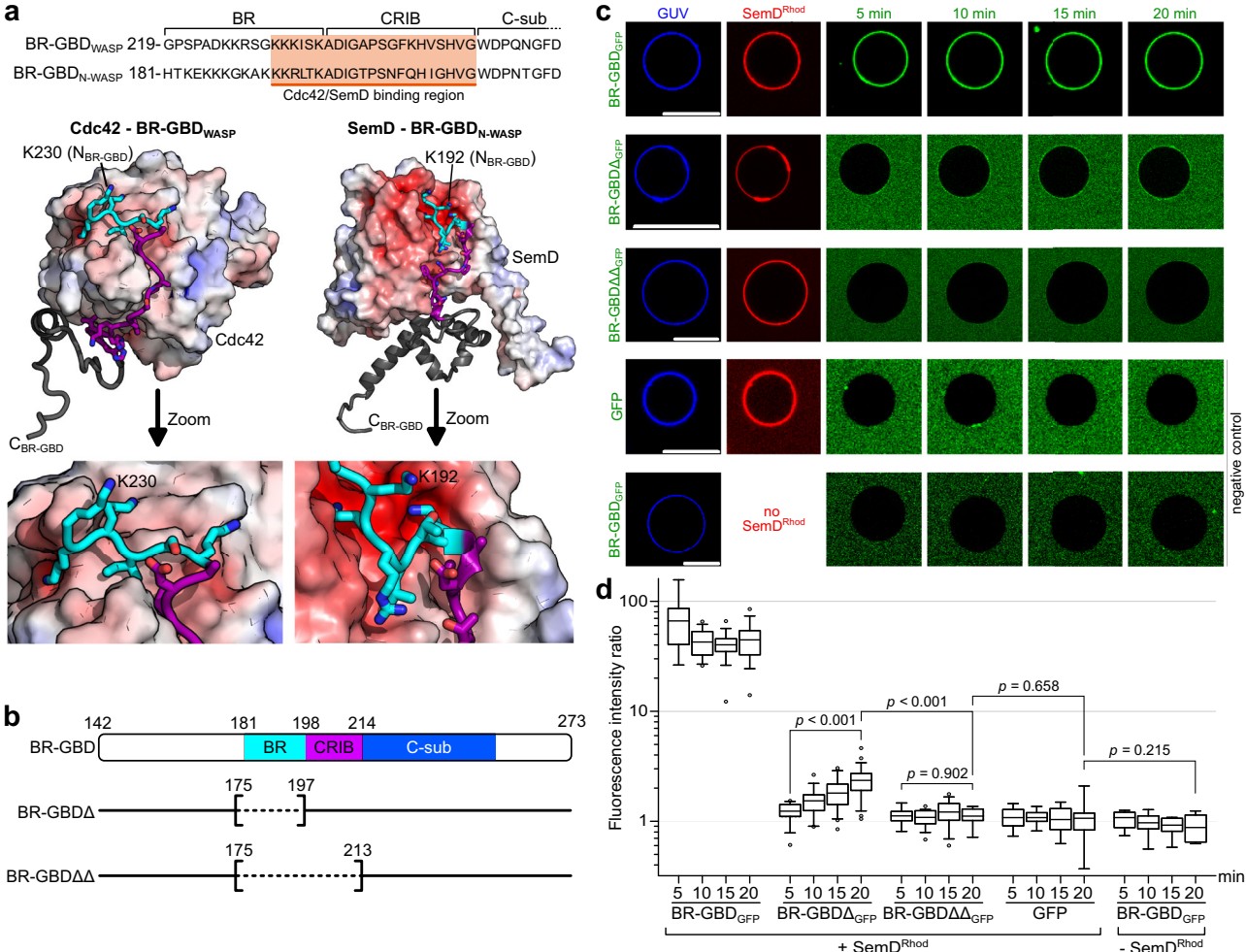

**Fig. 3 | SemD recruits the BR-GBD region of N-WASP to membrane vesicles.**
**a** Amino acid sequence of WASP$_{219-259}$ (*human*) and N-WASP$_{181-221}$ (*Rattus norvegicus*) (of which the latter is identical to N-WASP$_{184-224}$ from *human*) (*top*). The orange box shows the sequence involved in binding to Cdc42$_{GTP}$ and SemD, respectively. The lower panel shows the structures of Cdc42$_{GTP}$ in complex with BR-GBD$_{WASP}$ (PDB: 1CEE[37]) and SemDΔAPH bound to BR-GBD$_{N-WASP}$. Cdc42$_{GTP}$ binds to the positively charged C-terminal KKK$_{230-232}$ motif of the BR from WASP and embeds the CRIB domain in the less charged binding groove. The same binding mechanism is observed for SemD, in which a negatively charged patch engages with the positively charged KKR$_{192-194}$ motif within the N-WASP BR domain and inserts the subsequent N-WASP CRIB domain into the SemD binding groove. **b** Schematic representation of BR-GBD and the deletion variants BR-GBDΔ (lacking the BR$_{175-197}$ domain) and BR-GBDΔΔ (lacking the BR$_{175-197}$ and the CRIB$_{198-213}$ domains). Dashed

lines in square brackets mark the deleted protein regions. The first and last deleted amino acids are indicated. **c** Confocal images of PS-containing GUVs with rhodamine-labelled SemD (SemD$^{Rhod}$) and BR-GBD variants fused to GFP (BR-GBD$_{GFP}$). (scale bars 10 μm). **d** Quantification of bound protein to the GUV membrane based on the ratio of the maximal fluorescence at the perimeter of the GUV to the average background fluorescence outside the GUV. For each variant and time point, the fluorescence intensity ratio was calculated for up to 22 independent GUVs. The data are represented as boxplots with whiskers. The boxes are limited by the 25th and 75th percentile, including 50% of the data. The centre line shows the mean score. Whiskers denote 5–95% of all data and outliers are shown as grey dots. For comparing two groups, an unpaired, two-sided Student's *t* test was used. The data are representative for two independent data sets, both yielding similar results. Source data of both data sets are provided as Source Data file.

---

timescale (Fig. 4c). Collectively, these data show that SemD has a stronger binding capacity for N-WASP than active Cdc42$_{GTP}$, and is able to displace Cdc42$_{GTP}$ from the Cdc42$_{GTP}$ – N-WASP complex. Thus, during a *Cpn* infection, the locally secreted and PM-bound SemD underneath the invading EB most probably binds and activates cytosolic as well as Cdc42$_{GTP}$-bound N-WASP to initiate the branched F-actin mesh required for EB internalisation.

The preference of SemD for N-WASP over the physiological N-WASP activator Cdc42$_{GTP}$ raises the question of whether SemD is a specific activator of N-WASP or can also activate other Cdc42 target effectors. Cdc42 has been implicated as a key regulator of F-actin reorganisation, e.g. via activation of WASP/N-WASP to generate branched F-actin structures, as well as Formins, which play a critical role in nucleating actin filaments and promoting their elongation, thus influencing a large number of major cellular processes, involving actin dynamics such as cell

motility, cell division and intracellular transport[41]. Formins are auto-inhibited and require binding of active Cdc42$_{GTP}$ to the Formin GTPase binding domain for activation[42]. In in vitro pulldown assays, we tested whether SemD interacts with the mammalian FMNL2. We used recombinant FMNL2$_{GST}$ as bait and tested the binding of SemDΔAPH$_{His}$ and Cdc42$_{GppNHp}$ to it by analysing the FT and elution fractions (Fig. 4e). As expected, Cdc42$_{GppNHp}$ binds to FMNL2$_{GST}$, reaching an Elution:FT ratio significantly higher than the negative control with GST only (Fig. 4f). Interestingly, incubation of SemDΔAPH with GST or FMNL2$_{GST}$ showed no significant difference (Fig. 4e, f and Supplementary Fig. 4c, d). Thus, SemDΔAPH does not bind to FMNL2$_{GST}$, indicating that SemD specifically activates N-WASP and is not a general activator of Cdc42$_{GTP}$-dependent effector proteins, such as FMNL2.

Taken together, these data show that SemD copies Cdc42$_{GTP}$ at the EB entry side for N-WASP recruitment and activation.

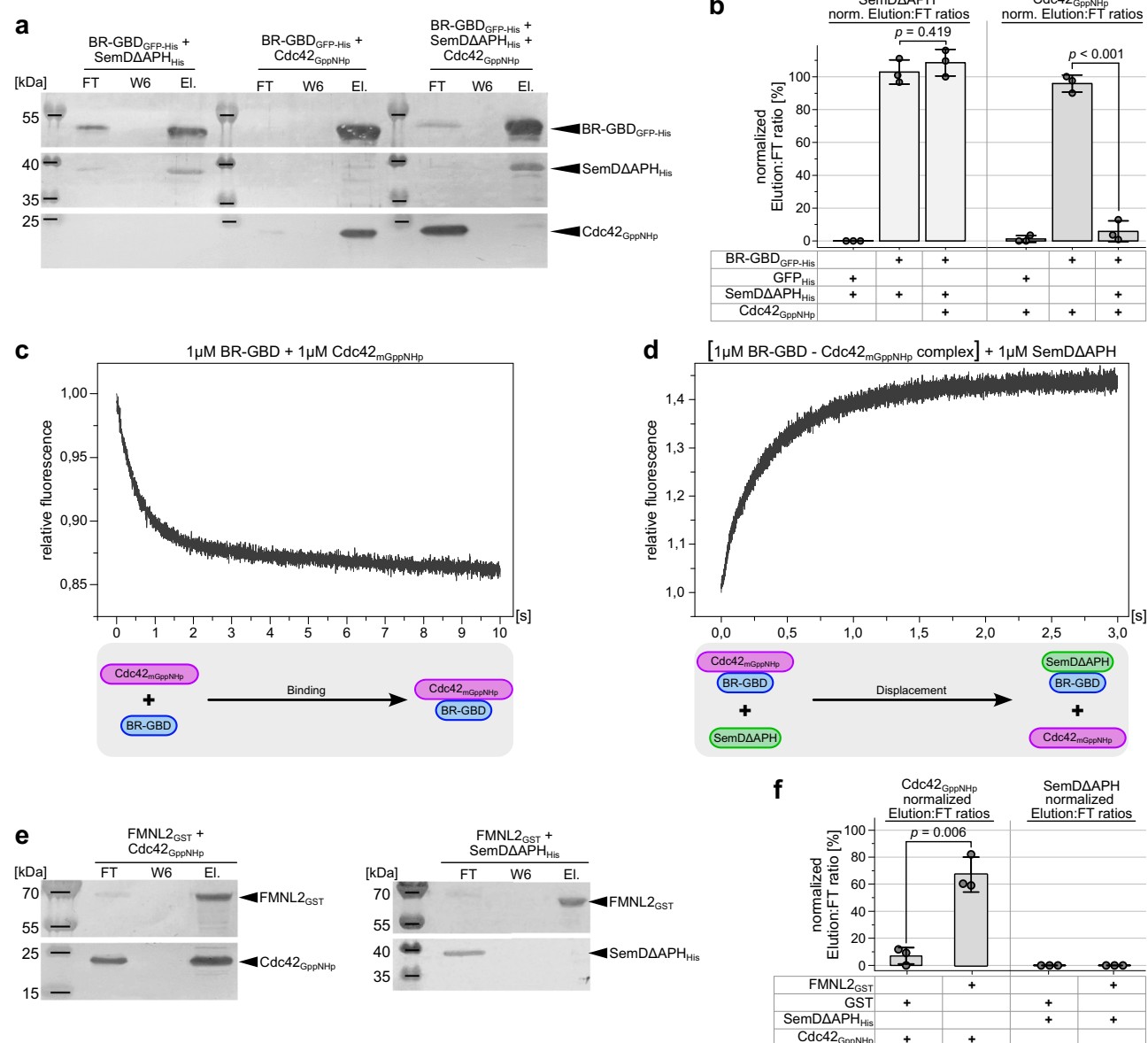

**Fig. 4 | Relative to Cdc42$_{GppNHp}$, SemDΔAPH displays enhanced and specific binding to N-WASP. a** GFP Trap® Pulldown experiments using equimolar ratios of purified recombinant SemDΔAPH$_{His}$, and active Cdc42 bound to the non-hydrolysable GTP analogue (Cdc42$_{GppNHp}$), were used to test their respective binding to BR-GBD$_{GFP-His}$. Complex formation between BR-GBD$_{GFP-His}$ and SemDΔAPH$_{His}$ or Cdc42$_{GppNHp}$ served as positive controls. Flow through (FT), wash 6 (W6) and elution (El.) fractions were analysed by SDS/PAGE and probed with anti-His (SemDΔAPH$_{His}$ and BR-GBD$_{GFP-His}$) and anti-Cdc42 (Cdc42) antibodies. Pulldown experiments were repeated three times with similar results. Replicates and negative controls are provided in Supplementary Fig. 4a and as Source Data to Fig. 4. **b** Quantification of western blotting in **a** is described in methods. Normalised (norm.) data are displayed as mean ± s.d. ($n = 3$ biologically independent experiments). Unpaired, two-sided Student's $t$-test was used to compare two groups. **c**, **d** Stopped-Flow experiments used to test the binding of equimolar ratios of BR-

GBD$_{His}$ and Cdc42$_{mGppNHp}$ (**c**), as well as the displacement of Cdc42$_{mGppNHp}$ from BR-GBD$_{His}$ by the addition of an equimolar amount of SemDΔAPH$_{His}$ (**d**). The lower panels schematically indicate the relevant interactions. **e** GST-Agarose pulldown experiments used to probe the interactions of Formin L2$_{GST}$ (FMNL2$_{GST}$) with Cdc42$_{GppNHp}$ and SemDΔAPH$_{His}$ with Cdc42$_{GppNHp}$. Complex formation between Cdc42$_{GppNHp}$ and FMNL2$_{GST}$ served as a positive control. Flow through (FT), wash 6 (W6) and elution (El.) fractions were analysed by SDS/PAGE and probed with anti-His (SemDΔAPH$_{His}$), anti-GST (FMNL2$_{GST}$) and anti-Cdc42 (Cdc42) antibodies. Pulldown experiments were repeated three times with similar results. Replicates and negative controls are provided in Supplementary Fig. 4b, c and as Source Data to Fig. 4. **f** Quantification of western blotting in **e** is described in methods. Normalised (norm.) data are displayed as mean ± s.d. ($n = 3$ biologically independent experiments). Unpaired, two-sided Student's $t$ test was used to compare two groups.

## The SH3 domain of SNX9 stabilises the PRD1 in SemD

The SemD – N-WASP structure also revealed that the PRD1 and PRD2 domains of SemD, which are required for binding of the SH3 domain of Pacsin 2/3 or SNX9, are located on the flexible N-terminus of SemD, and not in its core region. To ascertain whether the interaction of SemD with SNX9 affects the structure of SemD and whether simultaneous binding of SNX9 and N-WASP to SemD is

structurally feasible, we first examined the interaction between SemD and SNX9.

For this purpose, SemD, and a point mutated version ($_{mut}$SemD), in which the 12 proline and arginine residues in the PRD1 and PRD2 motifs were replaced by valine and alanine residues, respectively, were tested for interaction with the SH3 domain of SNX9 (Fig. 5a). Using PS-containing GUVs, that mimic the inner leaflet of the PM, we analysed

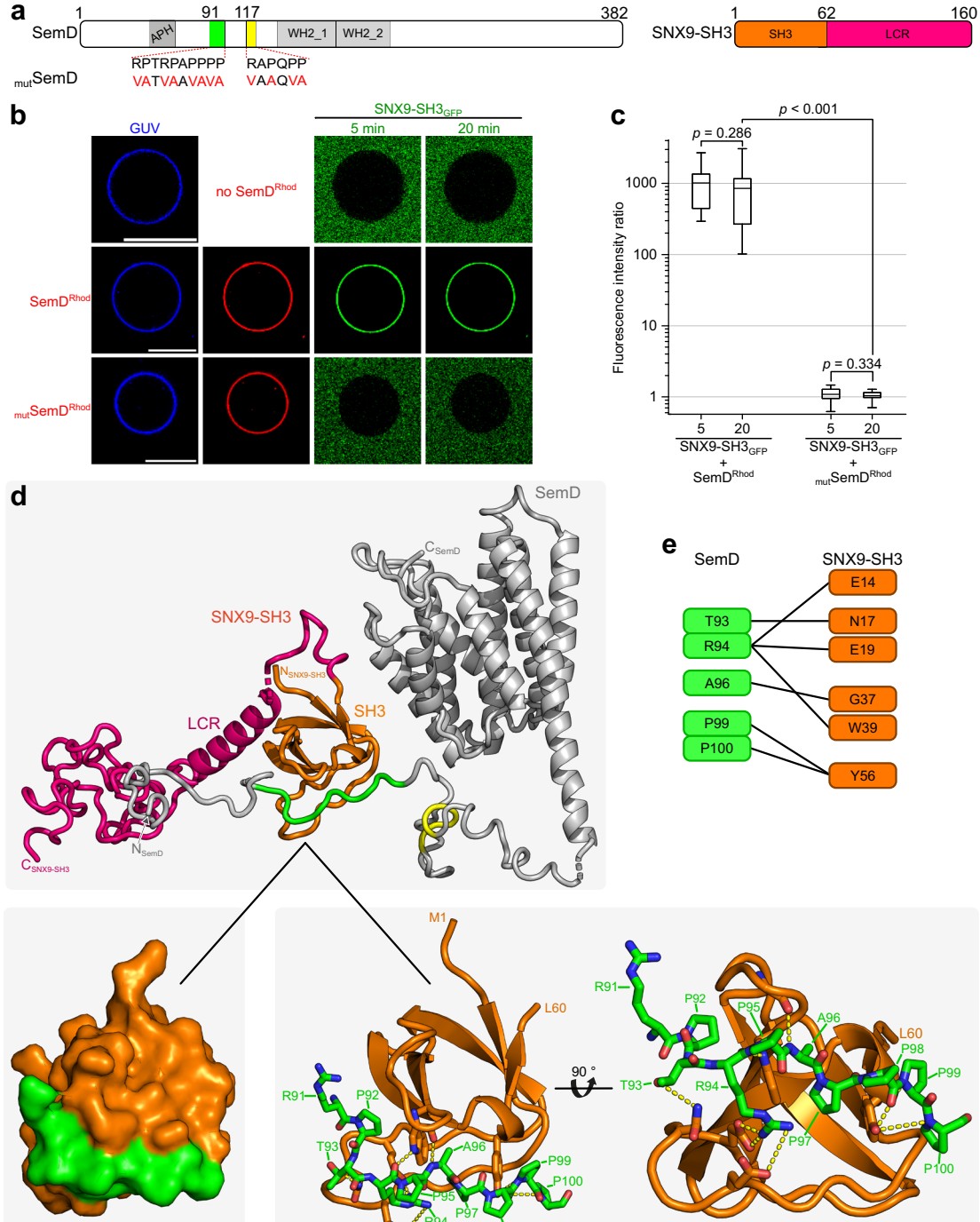

**Fig. 5 | SemD binds to SNX9-SH3 PRD1. a** (left) Schematic representation of SemD with PRD1 and PRD2 in green and yellow, respectively. Mutations for SemD_mut are indicated by the amino acids in red. (right) Schematic representation of SNX9-SH3 with the SH3 in orange and the low complexity region (LCR) in pink. **b** Confocal images of PS-containing GUVs with labelled SemD or SemD_mut and SNX9-SH3_GFP. (scale bars 10 μm). **c** Quantification of bound protein to the GUV membrane based on the ratio of the maximal fluorescence at the perimeter of the GUV to the average background fluorescence outside the GUV. For each variant and time point, the fluorescence intensity ratio was calculated for up to four independent GUVs. The data are represented as boxplots with whiskers. The boxes are limited by the 25th

and 75th percentile, including 50 % of the data. The centre line shows the mean score. Whiskers denote 5–95% of all data and outliers are shown as grey dots. For comparing two groups, an unpaired, two-sided Student's *t* test was used. The data are representative for three independent data sets, all yielding similar results. Source data of all data sets are provided as Source Data file. **d** Cartoon model of SemDΔAPH (grey) with SNX9-SH3 (SH3 in orange) as determined by SAXS using the programme CORAL. SNX9-SH3 (orange) binds to PRD1 (green) of SemDΔAPH. (zoom in) The interactions between the two domains are displayed. **e** Proposed details of the interactions between the two domains, based on the model shown in **d**.

the ability of membrane-bound SemD^Rhod and _mutSemD^Rhod to recruit SNX9-SH3_GFP (Fig. 5b). SemD^Rhod immediately bound to PS-GUVs and rapidly recruited SNX9-SH3_GFP (which does not bind to GUVs on its own), thus confirming published data[12] (Fig. 5b, c). Quantification

revealed essentially immediate saturation of SNX9-SH3_GFP binding to membrane-bound SemD^Rhod with no further increase over the next 20 min. Strikingly, _mutSemD^Rhod was unable to recruit SNX9-SH3_GFP to GUVs at all (Fig. 5b, c). This complete loss of the ability of _mutSemD^Rhod

to recruit SNX9-SH3$_{GFP}$ clearly indicates that the PRD1 and/or PRD2 domain(s) are responsible for SNX9-SH3 binding. To analyse the effect of binding on the SemD structure, we set out to characterise the interaction on a structural level. Thus, we expressed and purified recombinant SemDΔAPH and SNX9-SH3, allowed for complex formation and purified the resulting complex via SEC (Supplementary Fig. 5). The isolated complex was then analysed by SAXS (Fig. 5d).

SAXS analysis of SNX9-SH3 apo indicated that the protein is found as a monomer in solution with a structured core, and a flexible region indicated via the Kratky plot (Supplementary Fig. 6 and Supplementary Table 2). We initially modelled the SemDΔAPH – SNX9-SH3 complex with AlphaFold2[43] to get information about the binding interface. It is predicted that the SH3 domain of SNX9 interacts with the PRD1 in SemD (Fig. 5d and Supplementary Fig. 7, Supplementary Table 2). The N-terminal part of the SemDΔAPH AlphaFold2 prediction showed very low values of the predicted local distance difference test (pLDDT <30), in line with our conclusion of the flexibility. The only exceptions are the amino acids responsible for the interaction with the SH3 domain from SNX9-SH3 (pLDDT values ~80). The SNX9-SH3 prediction showed only for the N-terminal part including the SH3 domain high pLDDT values (between 60–80) and the remaining C-terminal part remains unclear (pLDDT values < 30, Supplementary Fig. 7b). To prove the resulting AlphaFold2 complex model, we calculated the theoretical scattering pattern of this model and compared it with the experimental scattering data of the SemDΔAPH – SNX9-SH3 complex in solution. The resulting CRYSOL fit offered a $\chi^2$ value of 8.49 and showed a high mismatch in the low s region (Supplementary Fig. 7c). This indicates that even the modelled complex contains all residues, and that the orientation of the domains/tails are not in line with the in-solution behaviour. The SAXS analysis of the SemDΔAPH – SNX9-SH3 complex confirmed a stoichiometry of 1:1, based on the molecular weight (Supplementary Table 2). We used the SemDΔAPH crystal structure and the binding interface of SH3 with PRD1 in SemD predicted by the AlphaFold2 model as a starting point for our modelling. The remaining flexible extensions were then remodelled with CORAL to better describe the in-solution behaviour of the SemDΔAPH – SNX9-SH3 complex (Supplementary Table 2). The resulting best-fit model of the remodelled SemDΔAPH – SNX9-SH3 complex is shown in Fig. 5d (an overlay of independent CORAL models is shown in Supplementary Fig. 7h). Based on our modelling, six residues of the SH3 domain bind to five specific residues of SemD PRD1 (Fig. 5d, e). Thus, PRD2 is not involved in direct contact with the SH3 of SNX9, in agreement with previous results obtained by Spona et al.[12], in which pulldown of SemD lacking the PRD1 showed no binding to SNX9.

Furthermore, structural analysis of the modelled complex showed that SH3 binding to PRD1 does not disrupt the conformation of the SemD core region, and therefore does not affect its ability to bind N-WASP. Indeed, the binding sites for host proteins on SemD are separated by flexible linkers, which minimise steric hindrance and allow the individual domains to be separately targeted in the 3D space.

### SemD binds simultaneously to several host partners

Our findings thus far indicate that SemD contains spatially separated binding sites that are connected by highly flexible linker regions, which sterically allow simultaneous interactions with the PM, N-WASP and SNX9. To assess the potential concurrent binding of these interaction partners, we mixed and incubated recombinantly expressed SemDΔAPH, SNX9-SH3 and BR-GBD, and analysed the composition of the complex in the sample using SEC. Indeed, the resulting chromatogram showed one main peak, eluting at 9.9 ml, that contained all three proteins, as evidenced by SDS/PAGE analysis (Fig. 6a).

To set these findings in context with membrane-bound SemD, we used our GUV model system to ascertain whether such a complex is formed under these conditions. After allowing SemD$^{Rhod}$ to bind to PS-containing GUVs, we added a three-fold molar excess (relative to

SemD) of either DyLight 650 labelled SNX9-SH3 (SNX9-SH3$^{DyLight\,650}$) or BR-GBD$_{GFP}$, to fully saturate SemD$^{Rhod}$, which forms a 1:1 complex with SNX9-SH3 and with BR-GBD, respectively. We then added the second binding partner in an equimolar ratio to SemD$^{Rhod}$. As expected, even after the initial saturation of SemD$^{Rhod}$ with one of the binding partners, the second partner was immediately recruited by SemD$^{Rhod}$ (Fig. 6b). This effect does not depend on either the sequence of addition or the quantity of the introduced binding partner, indicating that a stable complex with both partners can be consistently assembled on the target membrane.

By combining our X-ray crystallographic and SAXS data for SemDΔAPH – BR-GBD and SemDΔAPH – SNX9-SH3, we were able to develop a potential 3D model for the complex that includes all three interaction partners (Fig. 6c), which confirmed that simultaneous interactions with SNX9-SH3 and BR-GBD are sterically possible. However, the exact arrangement of the individual binding sites remains elusive owing to the flexibility of the linker regions connecting the individual binding sites in SemD. Hence, our data indicate that PM-bound SemD can simultaneously recruit the host endocytic proteins SNX9 and N-WASP using spatially separated binding domains.

## Discussion

As an obligate intracellular pathogen, *Chlamydia pneumoniae* interacts with host-cell proteins that ensure its survival and propagation. Perhaps the most critical stage in its replication cycle is its entry into a host cell.

For internalisation, the infectious EB (diameter 300-400 nm) requires co-option of the host's endocytic machinery to form a membrane-enclosed vesicle that is some 60 times larger in volume and 16 times larger in surface area than a classical endocytic vesicle (diameter 100 nm)[10]. This requires extensive remodelling of the PM and diversion of the host's actin cytoskeleton to enable growth, maturation and closure of the vesicle. Effector proteins translocated into the host cell play a vital role in these processes. The early secreted *Cpn* effector protein SemD binds to the inner leaflet of the PM below the invading EB and directly recruits G-actin and the essential endocytic proteins SNX9 and N-WASP[12].

Our structural study reveals that SemD interacts with host proteins via binding domains that are connected by intrinsically disordered linker sequences. This highly flexible arrangement facilitates simultaneous binding of several host endocytic proteins and modulation of the host's PM (Fig. 7). The precise contribution of these complexes to infection in vivo remains to be established, further complicated by the absence of a method for generating genetically manipulated *Cpn* strains.

We have shown in this study that SemD uses its C-terminal rigid core to bind and activate the actin nucleation- and branching-promoting factor N-WASP by structurally and functionally mimicking the normal role of the endogenous N-WASP activator Cdc42$_{GTP}$. Our structural and biochemical data further reveal that SemD provides the negatively charged patch and the binding groove required for selective binding of the positively charged C-terminal part of the BR and the CRIB domains of N-WASP, respectively, thus mimicking Cdc42$_{GTP}$-mediated N-WASP activation (Figs. 3a and 7). These interactions release the VCA domain of the autoinhibited N-WASP, which stimulates the Arp2/3 complex, thus promoting actin nucleation and branching. Intriguingly, the K$_{193}$R$_{194}$K$_{197}$ (KRK) motif in the C-terminal part of the BR region is responsible for both of these contacts with SemD (Fig. 2) and for binding to Cdc42$_{GTP}$, since a 9 aa deletion within the BR, that leaves the KRK motif intact, does not affect binding of the N-WASP mutant to Cdc42$_{GTP}$[25]. Mimicking of the endogenous Cdc42$_{GTP}$ protein, which is involved in many different cellular processes, requires specific activation of N-WASP by SemD. Based on our structural analysis, this occurs via the interaction of the KRK motif within the C-terminal BR region of N-WASP with a negatively charged patch on

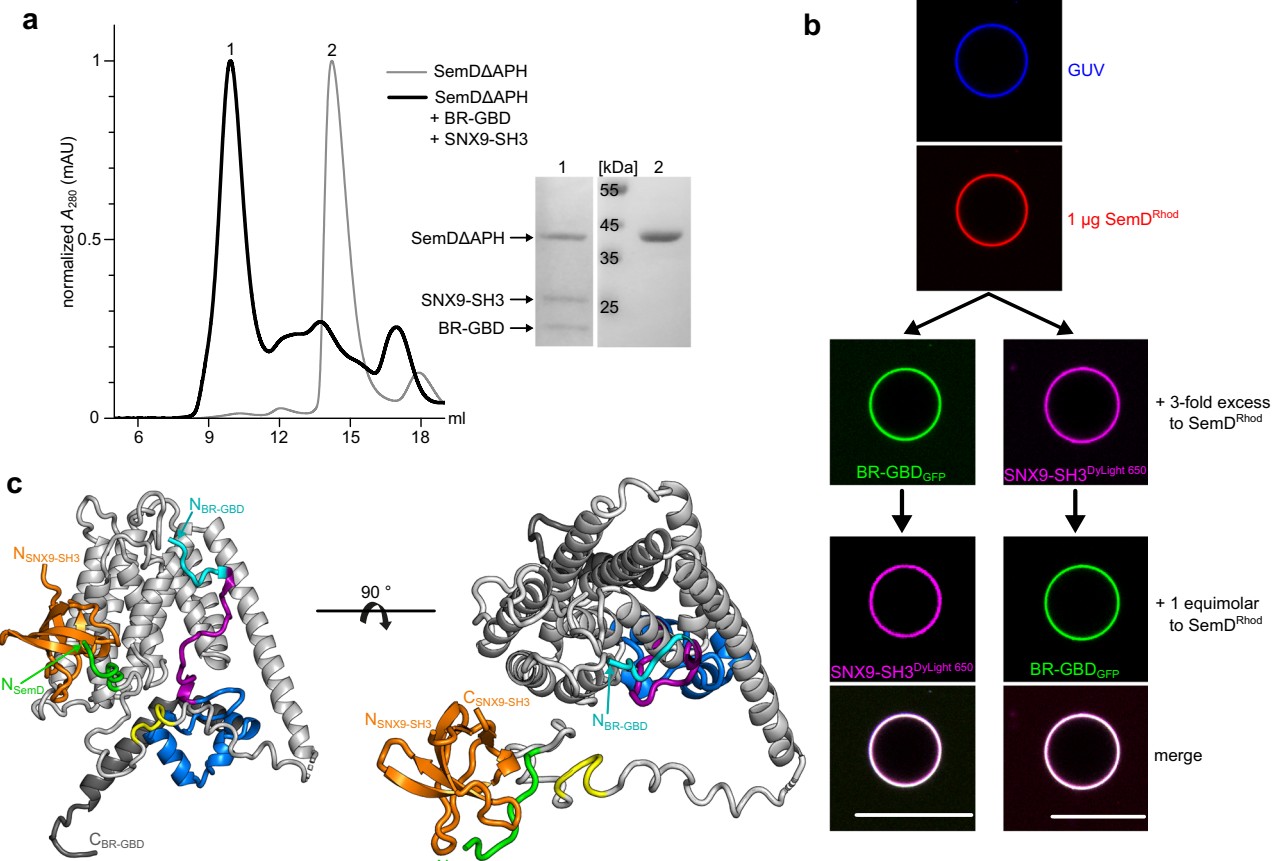

**Fig. 6 | SemD simultaneously interacts with various binding partners. a** SEC chromatograms of the complex composed of SemDΔAPH, BR-GBD and SNX9-SH3 (black), or SemDΔAPH alone (grey). The absorbance at 280 nm was normalised for the maximal absolute absorbance of the individual sample. The chromatogram of the complex revealed a major peak eluting at 9.9 ml (peak 1), while SemDΔAPH alone elutes at 14.2 ml (peak 2). The protein compositions of peaks 1 and 2 were analysed on an SDS gel (right) after staining with Coomassie brilliant blue. Lanes 1 and 2 were loaded with samples of the indicated peaks (*n* = 1). **b** Confocal images of PS-containing GUVs incubated with SemD$^{Rhod}$. A three-fold excess of either BR-GBD$_{GFP}$ (*n* = 4) or labelled SNX9-SH3 (SNX9-SH3$^{DyLight 650}$) (*n* = 6) was added, before

the third binding partner was added in an equimolar ratio to SemD (scale bars 10 μm). **c** The structures of SNX9-SH3 and BR-GBD obtained by SAXS overlaid on SemDΔAPH. The nine core helices of SemDΔAPH are depicted in grey and the PRD1 and PRD2 in green and yellow, respectively. BR-GBD is depicted in cyan, magenta and blue, in accordance with the colour scheme in Fig. 2, and the depiction of the SNX9-SH3 domain in orange follows the colour scheme used in Fig. 5. Note that the three-dimensional orientation of the bound SH3 domain towards the nine-helix core might be different, owing to the presence of the flexible linker in between the two. Right panel: 90° rotation.

SemD, which is much larger than that found on Cdc42 (Fig. 3a) and involves negatively charged amino acids on four different helices (α1, α2, α5 and α9) of the SemD's rigid core (Fig. 2c). Indeed, membrane-bound SemD recruits the BR-GBD segment more than 18-fold more efficiently than the BR-GBDΔ mutant, which lacks residues 181 to 197 including the KRK motif (Fig. 3). Comparison of the rigid core structure of SemD alone and when bound to the BR-GBD fragment reveals almost identical conformations, suggesting that SemD serves as a stable platform for BR-GBD, thus maximising the chances for fast recruitment via electrostatic interactions. Our pulldown and stopped-flow experiments indicate that SemD binds N-WASP much more tightly than active Cdc42$_{GTP}$ does, and indeed SemD can displace Cdc42$_{GTP}$ from the Cdc42$_{GTP}$ – N-WASP complex (Fig. 4a–d). Thus, during a *Cpn* infection, SemD is secreted via the T3SS by the adhering EB, interacts with the cytosolic leaflet of the PM and recruits and activates cytosolic, autoinhibited N-WASP, but might also dislodge N-WASP from Cdc42$_{GTP}$ – N-WASP complexes. The strong binding of SemD to N-WASP probably accounts for the efficiency with which the locally PM-bound SemD recruits N-WASP to establish the branched F-actin mesh required for EB internalisation. Moreover, *Cpn* has maximised this actin-branching process by evolving a SemD protein, which according to our data does not bind to FMNL2, which is also activated

by Cdc42$_{GTP}$ (Fig. 4e−f) and nucleates and elongates unbranched actin filaments at the barbed end[41]. Comparison of Cdc42$_{GTP}$ – N-WASP and SemD – N-WASP (Fig. 3a) with the Cdc42$_{GTP}$ – FMNL2 structure[44] reveals remarkable differences. N-WASP strongly interacts via its positively charged residues in the BR and the CRIB with the negatively charged patch on the front of Cdc42$_{GTP}$ and SemD, respectively. Conversely, FMNL2 interacts with Cdc42$_{GTP}$ via multiple hydrophobic and polar contacts formed between all five armadillo repeats of FMNL2 and the two switch regions of Cdc42. These differences probably account for the inability of SemD to bind FMNL2[44].

Thus, SemD is not only a very efficient activator of N-WASP, but is likely to be restricted in its activity to that protein. This would ensure that the limited numbers of SemD molecules secreted by the invading *Cpn* are fully available for this process. For actin nucleation and elongation of unbranched actin filaments *Cpn* secretes within the first 15 min of infection the soluble effector protein CPn0572, which belongs to the TarP protein family[45].

The SemD-mediated local reorganisation of the actin network is probably transient and short-lived, until bacterial entry has succeeded. In *Salmonella*, following host-cell entry, the architecture of the cytoskeleton is restored by, for example, the bacterial GTPase-activating protein SptP, which reverses the activation of Rac1

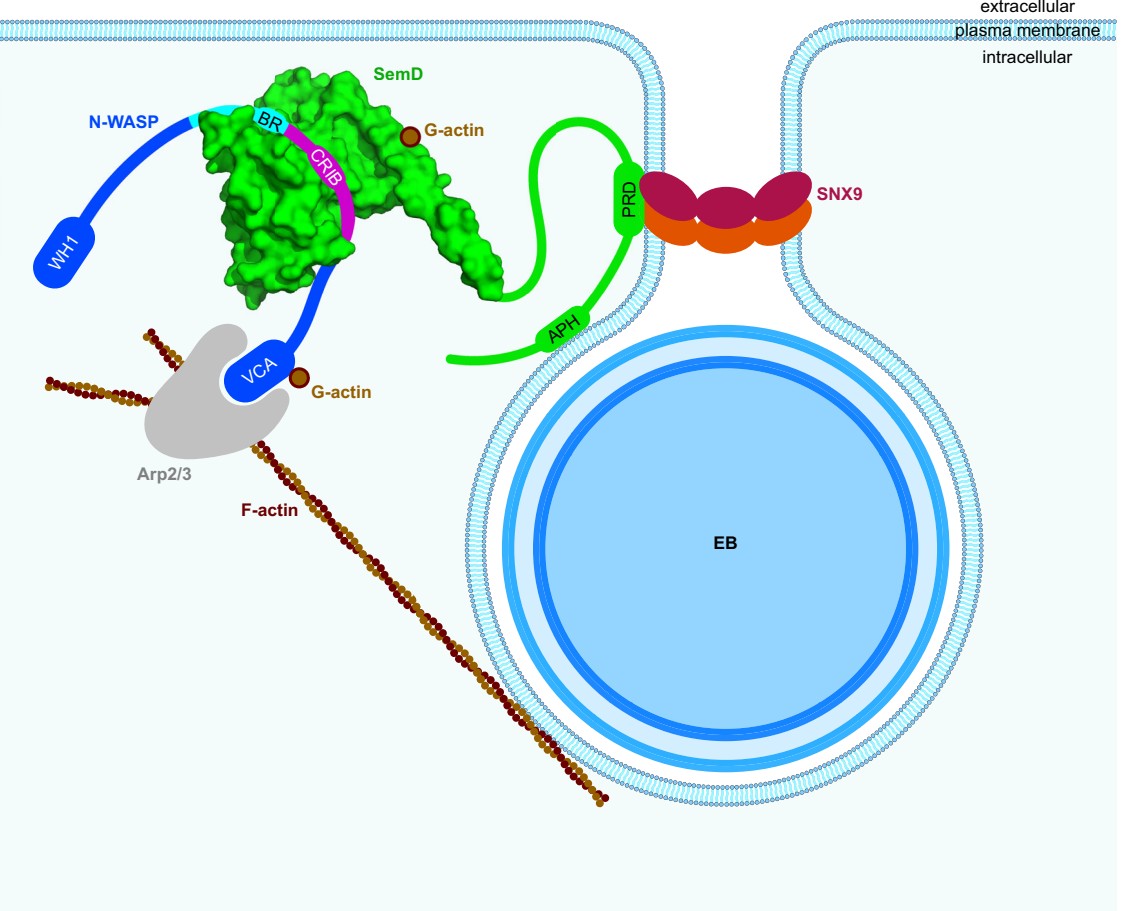

**Fig. 7 | A chlamydial effector exploits structural and functional mimicry to manipulate the host endocytic machinery.** The *Cpn* elementary body (EB) secrets SemD into the host cell, which binds to the inner leaflet of the plasma membrane. There, SemD recruits, binds and activates N-WASP by structurally and functionally mimicking the Cdc42$_{GTP}$ activation mechanism. SemD interacts with the C-terminal, positively charged amino acids of the N-WASP BR domain and further, the CRIB domain binds into the SemD binding groove. This then leads to the release of N-WASP from its auto-inhibited state. SemD also binds to the SNX9-SH3 domain, which brings the SNX9-BAR domain closer to the membrane. This in turn induces membrane deformation and eventually leads to closure of the matured endocytic vesicle. Due to the arrangement of the individual binding domains, which are connected by flexible linker regions, the binding sites can be freely oriented in 3D space, thus minimising steric hindrance. This can explain why SemD is postulated to be capable of binding simultaneously to the PM, SNX9 and N-WASP in vivo and hijacking their functions to promote the growth and maturation of the endocytic vesicle.

and Cdc42[46]. It will be interesting to ascertain how this is achieved by *Chlamydiae*.

The SemD-mediated activation of N-WASP differs fundamentally from that triggered by other pathogens, which evolved effector proteins mimicking modulators of Cdc42 activity, such as GAPs, GEFs and GDIs, or utilise covalent modification of the Rho GTPase (see introduction). The SemD activity is also completely different from the function of the effector EspF$_U$ secreted by enterohaemorrhagic *Escherichia coli* (EHEC). NMR data have revealed that EspF$_U$ binds the GBD domain via a C-like motif (similar to that found within the VCA domain) that releases the endogenous VCA domain in autoinhibited N-WASP[35,47]. Interestingly, initial data suggest that the *Ctr* effector TmeA might activate N-WASP like EspF$_U$ does which would imply that *Cpn* has evolved a completely different mechanism for F-actin polymerisation and branching, possibly as an adaptation to the different target tissues involved (*Cpn*: lung epithelia; *Ctr*: eye + urogenital tract epithelia)[48,49].

SemD also binds the BAR-containing protein SNX9, which is required for membrane deformation and recruitment of dynamin, and eventually leads to the scission of the matured vesicle[50]. Our model analysis suggests that, in the flexible N-terminal half of SemD, which is separated by a linker sequence from the rigid core that mediates N-WASP interaction, five residues in the PRD1 domain interact with six residues in the β-sheet structure of the SNX9-SH3 domain (Figs. 5 and 7).

This binding mechanism is typical for proline-rich peptides that interact with SH3 domains, as has been shown for several other interaction partners (PDB: 1QWE, 2JMA, 2DRK, 2KXC). The predicted PRD2 domain is not involved in SH3 binding, as previously suggested by Spona et al.[12]. Thus, the amino acid sequences N- and C-terminal to PRD1 remain unstructured and may act as linkers that separate the PRD1 – SNX9-SH3 complex from the N-terminal membrane-binding domain APH and the C-terminal SemD core domain, which is involved in N-WASP binding (Figs. 6 and 7). Our structural model, based on the individual conformations of each protein pair (SemD + SNX9-SH3; SemD + N-WASP$_{BR-GBD}$), reveals that all protein interactions can occur simultaneously (Fig. 6c), and we have verified this by biochemical and membrane binding experiments that confirm concurrent recruitment of SNX9-SH3 and BR-GBD to GUV-bound SemD (Fig. 6b).

Helices α1 and α2 of SemD's rigid core carry the two predicted WH2 sequences essential for G-actin binding[12]. However, the stoichiometry of this interaction is not clear. Our SemD structure implies that WH2_1 on α-helix 1 is largely available for interaction with G-actin, while WH2_2 on α-helix 2 is not fully accessible (Fig. 1b, c), suggesting that WH2_1 might constitute the G-actin binding domain. Recruitment of G-actin by SemD increases the local G-actin concentration, which should promote formation of F-actin branches via the N-WASP-Arp2/3 pathway.

During evolution, *Cpn* has undergone a dramatic reduction in genome size to about 1 million bp in total. Consequently, many proteins must perform more than one task and our structural analysis reveals that this holds for SemD, an effector protein that is involved in the reshaping of membrane structure and actin cytoskeletal organisation during chlamydial endocytosis. Our data support a model in which a single PM-bound chlamydial effector protein, SemD, can simultaneously interact with several host proteins by separating the SemD binding domains with unstructured linker regions. Hence, the ability of SemD to mimic $Cdc42_{GTP}$ permits recruitment, binding and activation of the endocytic host protein N-WASP.

## Methods

### Antibodies and reagents

All lipids used in this study were obtained from Avanti Lipids and NHS-Rhodamine and DyLight650-NHS were sourced from Thermo Scientific. The primary antibody anti-penta-His (#34660, 1:2500) was purchased from Qiagen, anti-Cdc42 (#610929, 1:1000) was obtained from BD Transduction Laboratories and anti-GST (sc-374171, 1:500) was obtained from Santa Cruz Biotechnology. The secondary anti-mouse antibody coupled to alkaline phosphatase (#A3562, 1:30000) was purchased from Sigma-Aldrich.

### Cloning, protein expression and purification

Cloning steps were carried out by in vivo homologous recombination in *Saccharomyces cerevisiae*. *semD* constructs used in this study were amplified from synthetic *semD* DNA purchased from GenScript, which was codon optimised for *Escherichia coli* (*E. coli*) expression. The *SNX9-SH3* sequence was amplified from a sequence encoding mCherry-SNX9[13], the *BR-GBD* fragments were amplified from a sequence encoding GFP-N-WASP (Addgene, #47406, *Rattus norvegicus*; The N-WASP BR-GBD protein fragment from *R. norvegicus* and *human* differ by 3 amino acids located N-terminal to the BR domain outside of our co-crystal structure). The fragments were integrated either into pSL4 (generating C-terminal 10xHis fusions) or into pDS94 (generating C-terminal GFP-10xHis fusions) (Plasmid list in Supplementary Table 4). The plasmid encoding FMNL2 (S171DD) fused to GST has been published previously[44]. Expression of the His-tagged proteins was carried out in *E. coli* BL21 (DE3, Invitrogen), and expression of GST-tagged FMNL2 was carried out in *E. coli* Rosetta. His-tagged proteins were purified using Ni-NTA Agarose (Cube Biotech) and dialysed in phosphate-buffered saline (PBS) (10 mM $Na_2HPO_4$, 1.8 mM $KH_2PO_4$, 137 mM NaCl, 2.7 mM KCl, pH 8.5), apart from SemDΔAPH for which a pH of 6.0 was used. GST-tagged FMNL2 was purified using Glutathione Agarose (Thermo Scientific) and dialysed in buffer containing 10 mM Tris-HCl, 150 mM NaCl, pH 8.5.

The preparation of Cdc42 in complex with guanosine 5′-(βγ-imino)-triphosphate (GppNHp) and N-methyl-anthraniloyl-labelled GppNHp (mGppNHp) was carried out as described by Eberth and Ahmadian[51]. In brief, human *CDC42* was integrated into pGEX-4T-1 (generating a GST-fusion)[23] and expressed in *E. coli* Rosetta. GDP-bound GST-Cdc42 was purified using a Glutathione sepharose column (Pharmacia, Uppsala, Sweden) and the GST-tags were cleaved with thrombin at 4 °C overnight. Proteins were reapplied to Glutathione Sepharose and cleaved Cdc42 was collected in the flow through. Protein quality and concentration were assessed by SDS-PAGE and high-performance liquid chromatography (HPLC), utilising a Beckman Gold HPLC system with a reversed-phase C18 column. GDP-bound Cdc42 proteins were incubated with a 1.5-fold excess of GppNHp/mGppNHp, non-hydrolysable GTP analogues, and agarose bead-coupled alkaline phosphatase (0.1–1 U per mg of Cdc42) to degrade GDP to GMP and Pi, thus facilitating the replacement of GDP with GppNHp/mGppNHp. The course of the reaction was monitored via HPLC using a buffer containing 100 mM potassium phosphate (pH 6.5), 10 mM tetra-butylammonium bromide, and 7.5–25% acetonitrile. Upon complete

degradation of GDP, the samples were applied to prepacked NAP-5 columns to exchange the buffer for a fresh one devoid of free nucleotides. The concentration of nucleotide-bound Cdc42 was checked using the Bradford assay and HPLC to calculate the amount of active GppNHp-bound Cdc42. The proteins were then snap-frozen and stored at −80 °C for downstream analysis. Preparatory steps of $Cdc42_{GppNHp}$ are provided in Source Data.

### Size exclusion chromatography

SEC was performed on an ÄKTA™ pure 25 L (Cytiva). For purified proteins, a pre-equilibrated HiLoad 16/600 Superdex 200 pg column was used with a flow-rate of 0.8 ml/min, for pre-formed complexes, a pre-equilibrated Superdex 200 increase 10/300 GL column (Cytiva) was used with a flow rate of 0.5 ml/min. All runs were performed at 4 °C.

### Pulldown assays

Recombinant BR-GBD fused to GFP, or GFP alone, was mixed with an equimolar ratio of the test protein(s) and incubated for 5 min at RT. GFP Trap® agarose, preincubated in 3 % BSA, was added to the mixture, and binding was allowed to proceed for 30 min at 4 °C. After collection of the flow through, agarose was washed 6x with wash buffer (10 mM Tris-HCl, 200 mM NaCl, pH 8.5) and bound proteins were eluted by boiling the agarose in SDS sample buffer.

Recombinant FMNL2 fused to GST, or GST alone, was mixed with an equimolar ratio of the test protein(s) and incubated for 5 min at RT. Glutathione agarose was added to the mixture and binding was allowed for 30 min at 4 °C. After collecting the flow through, agarose was washed 6x with wash buffer (50 mM Tris-HCl, 150 mM NaCl, pH 8.5) and bound proteins were eluted by boiling the agarose in SDS sample buffer.

Individual steps were monitored by SDS/PAGE and immunoblot analysis, using specific primary and secondary antibodies.

### Western blot quantification

Band intensities of the Flow Through (FT) and elution (El.) fractions were determined using the software GelAnalyzer 23.1.1. Bands were semi-automatically defined. The Elution:FT ratio [%] was calculated by dividing the intensity of the eluate by the total intensity, i.e. FT plus eluate. The ratio was normalised to the Elution:FT ratio [%] of the bait protein used. Individual band intensities and uncropped western blots are displayed in Source Data.

$$\text{Elution : FT ratio[\%]} = \frac{\text{Elution}}{\text{Elution} + \text{FT}} \qquad (1)$$

$$\text{normalized elution : FT ratio [\%]} = \text{Elution : FT ratio} * \frac{1}{(\text{Elution : FT ratio})_{\text{bait}}} \qquad (2)$$

### Fluorescence stopped-flow spectrometry

Rapid fluorescence measurements were performed using a Hi-Tech Scientific stopped-flow spectrophotometer (Applied Photophysics SX20), as described by Hemsath et al.[23]. An excitation wavelength of 360 nm was used for N-methylanthraniloyl (m) derivatives of guanosine nucleotides in the stopped-flow analysis. Fluorescence detection was facilitated by a photomultiplier equipped with a cut-off filter to detect wavelengths above 408 nm. The association of N-WASP BR-GBD with mGppNHp-bound Cdc42 was measured using a buffer containing 30 mM Tris-HCl, 10 mM $K_2HPO_4/KH_2PO_4$ (pH 8.5), 5 mM $MgCl_2$ and 3 mM DTT at 25 °C. The experiment setup involved equimolar ratios of the proteins. The contents of one syringe containing 1 μM of mGppNHp-bound Cdc42 and a second syringe containing 1 μM N-WASP BR-GBD were rapidly mixed, and the change of relative

fluorescence was monitored in real-time. In a subsequent experiment, competition between SemDΔAPH and mGppNHp-bound Cdc42 was evaluated by rapidly mixing 1 μM of the pre-prepared complex of mGppNHp-bound Cdc42 and N-WASP BR-GBD with 1 μM of SemD-ΔAPH. The change in relative fluorescence was monitored in real-time.

## Preparation of giant unilamellar vesicles

GUVs were prepared as described previously[52]. Briefly, PS-containing GUVs were prepared by mixing 9.75 mol% DOPC, 25 mol% cholesterol, 0.25 mol% Marina Blue™ DHPE and 25 mol% DOPS. Lipid mixtures were prepared and added to a chamber built of ITO-coated slides (Präzisions Glas & Optik) which were glued together with Vitrex (Vitrex Medical). The resulting cavity was filled with 10 % sucrose solution and sealed with Vitrex. The slides were connected via clamps to a frequency generator and an alternating voltage of 2.0 Vp-p was applied at a frequency of 11 Hz. The GUVs were grown in the dark at room temperature for 2–3 h.

## Protein binding studies on giant unilamellar vesicles

For microscopic analyses, Angiogenesis μ-slides (Ibidi) were coated for 5–10 min at RT with 2 mg/ml β-casein (Merck) and washed three times with PBS. Then NHS-rhodamine-labelled recombinant SemD (1 μg) was mixed with 15 μl PBS and recombinant binding partner fused to GFP (1 μg) was added, together with 5 μl GUVs. For binding studies with three proteins, NHS-rhodamine-labelled recombinant SemD (1 μg) was mixed with recombinant BR-GBD fused to GFP (1-3 μg) and NHS-650–labelled recombinant SNX9-SH3 (1-3 μg) and 5 μl GUVs were added. The GUVs were allowed to settle down for 5 min at room temperature and then imaged for further 15 min at room temperature.

## Microscopy

General imaging was performed using an inverse Nikon TiE Live Cell Confocal C2plus equipped with a 100x TIRF objective and a C2 SH C2 Scanner. All images were generated with Nikon NIS Elements software and quantified using ImageJ.

## Fluorescence intensity ratio analysis

Acquired confocal GUV data were semi-automatically analysed using a self-written *fiji* macro to estimate signal accumulation at the perimeter of the GUV in relation to the surrounding medium. Multiple line selections were orthogonally placed at the GUV membrane, with the membrane placed in the middle. First, the macro plots a line-intensity profile for each selection with a given predefined linewidth (here: 5) in order to extract intensity data for the relevant signal channel and to store it in an array. The intensity of the signal surrounding the GUV is calculated as the mean intensity ($I_{out}$) of a predefined width (here: 1/5 of total profile length) at the front end of the line profile. Second, signal intensity peaks of the profile are identified by applying the built-in "array.findMaxima"-function with a given tolerance (here: 450), that returns peaks by default in descending significance order. The peak positions are checked, if they are located within a predefined width around the centre of the line profile (here: same as background width) and the intensity of the first, most significant peak ($I_{peak}$) within the limits, it is used for the following ratio calculation. Finally, the intensity ratio ($r_{int}$) is calculated as the quotient of signal intensity at the peak position ($I_{peak}$) divided by the mean signal intensity in the surrounding medium ($I_{out}$).

$$R_{int} = \frac{I_{peak}}{I_{out}} \qquad (3)$$

## Statistical analysis and data representation

Graphs were prepared using OriginPro v.2021b (OriginLab). For the comparison of two groups, an unpaired, two-sided Student's *t*-test was used. A *p*-value of less than 0.01 was considered as statistically significant. Images were prepared using the open-source software Inkscape (www.inkscape.org).

## Structure determination via crystallisation

SemDΔAPH, either alone or in a complex with BR-GBD, was crystallised by sitting-drop vapour-diffusion in PBS at pH 6 (SemDΔAPH) or pH 8.5 (SemDΔAPH + BR-GBD) at 12 °C and at concentrations of 24 and 10 mg/ml, respectively. 0.1 μl were mixed with 0.1 μl of reservoir solution consisting of 0.1 M Citric acid (pH 2.5), 20% (w/v) PEG 6000 (pH 4) for SemDΔAPH and 0.1 M ammonium formate, 0.1 M MES (pH 6.2), 25% v/v PEG 400 for SemDΔAPH + BR-GBD. Crystals formed after 12-24 h (SemDΔAPH) or 5 d (SemDΔAPH + BR-GBD) were harvested and cryo-protected with mineral oil followed by flash-freezing in liquid nitrogen. Diffraction data were collected at −173 °C (100 K) at beamline P13 (DESY, Hamburg, Germany) using a 0.9763 Å wavelength for SemD-ΔAPH or at beamline ID30A-3 (ESRF, Grenoble, France) using a 0.9677 Å wavelength for SemDΔAPH + BR-GBD. Data reduction was performed using XDS[53] and Aimless[54] from the CCP4 Suite[55]. The structure was solved via molecular replacement with Phaser[56] using an AlphaFold[57] model (SemDΔAPH) or the apo structure (SemDΔAPH + BR-GBD) as search model. The initial model was refined alternating cycles of manual model building in COOT[58,59] and automatic refinement using Phenix[60] v.1.19.2. Data collection and refinement statistics are reported in Supplementary Table 1. In the SemDΔAPH + BR-GBD structure one amino acid, Glu207, was found not to obey the Ramachandran rule, and is positioned in the disallowed region. This residue is involved in a crystal contact.

## SAXS measurement

SEC-SAXS data were collected on the P12 beamline (PETRA III, DESY Hamburg[61]). The sample-to-detector distance of the P12 beamline was 3.00 m, resulting in an achievable q-range of 0.03-0.07 nm-1. The measurements were performed at 20 °C with a protein concentration of 8 mg/ml for SemDΔAPH, 10 mg/ml for BR-GBD, 8 mg/ml for SemDΔAPH + BR-GBD and 3.3 mg/l for SemDΔAPH + SNX9-SH3. The SEC-SAXS runs were performed on a Superdex200 increase 10/300 GL column (100 μl injection volume, buffer: PBS pH 8.5 + 3 % glycerol) with a flow rate of 0.6 ml/min. 2400 frames were collected for each protein sample with an exposure time of 0.995 sec/frame. Data were collected on relative scale or absolute intensity against water.

All programmes used for data processing were part of the ATSAS Software package (Version 3.0.5)[62]. Primary data reduction was performed with the programmes CHROMIXS[63] and PRIMUS[64]. With the Guinier approximation[65], the forward scattering $I(0)$ and the radius of gyration ($R_g$) were determined. The programme GNOM[66] was used to estimate the maximum particle dimension ($D_{max}$) with the pair-distribution function $p(r)$. The rigid body results from the crystal structure were used as a starting template to complete the structures of SemDΔAPH and BR-GBD (flexible N- and C-terminal parts were remodelled) with the programme CORAL[67]. The flexibility ensemble analysis of the SemDΔAPH + BR-GBD complex was done with EOM[68,69], based on the solved crystal structure and completed with the missing amino acids. The SemDΔAPH + SNX9-SH3 complex docking was done with CORAL[67], based on the solved SemDΔAPH structure and an AlphaFold2[43,57] prediction of the interaction site from the SH3 domain with the flexible SemDΔAPH tail.

## Reporting summary

Further information on research design is available in the Nature Portfolio Reporting Summary linked to this article.

# Data availability

We uploaded the SAXS data to the Small-Angle X-ray Scattering Biological Data Bank (SASBDB)[70], with the following accession codes: SASDTQ5 (SemDΔAPH), SASDTR5 (BR-GBD), SASDTS5 (SNX9-SH3),

SASDTT5 (SemDΔAPH + SNX9-SH3) and SASDTU5 (SemDΔAPH + BR-GBD). The crystal structures were deposited in the Protein Data Bank (PDB) with the accession codes 8S5R (SemDΔAPH) and 8S5T (SemDΔAPH+ BR-GBD). Further, the cited structures in this paper can be found with the following accession codes: 1QWE (C-SRC SH3 + APP12). 2JMA (R21A Spc-SH3:P41 complex). 2DRK (SH3 + Acan125). 2KXC (IRTKS-SH3 + ExpFu-R47). 1CEE (Cdc42 + WASP). The authors declare that the data supporting the findings of this study are available within the paper and its extended data files. Data underlying Figs. 3d, 4a, b, e, f, 5c and 6a and Supp. Figs. 2a, 4a, b, c, d and 5a are provided as Source Data files. All other data are available from the corresponding author upon request. Source data are provided with this paper.

## Code availability

A custom code for Fiji 1.54 f used for the analysis of GUVs is available on https://github.com/SHaensch/2023_GUVQuant or https://doi.org/10.5281/zenodo.13165623.

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

## Acknowledgements

We acknowledge DESY (Hamburg, Germany), a member of the Helmholtz Association HGF, for the provision of experimental facilities. Parts of this research were carried out at PETRA III and we would like to thank Haydyn Mertens and Dmytro Soloviov (EMBL Hamburg) for their assistance in using beamline P12. We also acknowledge the European Synchrotron Radiation Facility (ESRF) for the provision of synchrotron radiation facilities and we would like to thank Petra Pernot, for assistance in using beamline BM29. The crystal datasets were collected at the P13 beamline at DESY Hamburg and ID30A-3 at the ESRF Grenoble. Here, we would like to thank Selina Storm and Igor Melnikov for their excellent support during data collection. We thank our reviewers for the constructive comments. F.K. was a scholarship holder of the Graduate School "Molecules of Infection IV (MOI IV)" funded by the Jürgen Manchot Foundation. The Center for Structural Studies is funded by the DFG (Grant Nos. 417919780, INST 208/761-1 FUGG, INST 208/868-1 FUGG and INST 208/740-1 FUGG to S.H.J.S.). We acknowledge grant support from the DFG to A.M. and M.R.A. (grant number: AH 92/8-3) and to J.H.H. (Project-ID 267205415) of CRC 1208.

## Author contributions

Conceptualisation: F.K., K.M. and J.H.H.; methodology F.K., K.M., M.R.A., A.M., S.H.J.S. and J.H.H.; investigation: F.K., V.A., J.R., A.P., D.S., S.H., A.M., M.R.A., S.H.J.S. and K.M., writing—original draft: F.K., K.M., S.H.J.S. and J.H.H.; writing—review & editing: F.K., K.M., S.H.J.S. and J.H.H; funding acquisition: J.H.H. and S.H.J.S.

## Funding

## Competing interests

The authors declare no competing interests.
