## [Peer Review File · Nature Communications]

The *Chlamydia pneumoniae* effector SemD exploits its host's endocytic machinery by structural and functional mimicryREVIEWER COMMENTS

Reviewer #1 (Remarks to the Author):

This study provides a very detailed biochemical and structural analysis of the SemD effector function and interaction with host cell partners. The manuscript is well written and the quality of the figures is excellent. The manuscript provide the crystal structure of SemD alone and in complex with N-WASP as well as some SAXS data on three different samples. The GUV binding data, together with mutational analysis, elegantly demonstrates the relevance of the biochemical information gained. I believe this study presents a very significant amount of novel information with some excellent biochemistry. Altogether, it provides detailed mechanistic insights into a single effector tailed to interact multiple cell partners for reprogramming the target cell.

Generally the interpretation of the data is good and the conclusion supported by the experimental results. However, the authors often refer to these models as a "SAXS structures" while SAXS was mainly used to model unstructured regions and to confirm in silico prediction. Indeed, the authors have used alpha fold but do not mention this in the result section of the manuscript and this is not appropriate and might even be misleading. Otherwise, the methodology is sound except for the crystallography data that requires significant improvement/modification or at least convincing explanations.

Major :

P7 : The sentence « Based on the information derived from SAXS, we added the missing N-terminal residues (aa 67-152) to complete the SemDDAPH structure and the resulting model indeed confirmed that these residues make up the IDR.

To draw this conclusion, the authors have used CORAL to model the N-terminal moiety. However the argument presented as such is circular given that the program CORAL models only flexible region. Similarly, Fig S1c as such is not informative because CORAL is made to fit best the SAXS curve so it seems a circular argument. Finally, CORAL can give more than one solution to fit the SAXS, often enabling to illustrate the flexibility. Fig S1c could be modified by showing the fit of the crystal structure AND the CORAL complete model and Fig 1 shows the different possible orientations of the N-terminal. This will allow the reader to understand how the modelling of the N-terminal part is required to fit the SAXS curve, hence providing a more realistic model of the protein. Similar issue occurs for modelling the complex of SemD/SNX9 (see below).

The crystal structure of SemDSAPH has been solved at 2.4 resolution.

- 1) First it is not clear why the resolutions has been cut at 2.4A while the I/sI is 5, R-pim is 0.15. From these numbers, it seems that the crystal was diffracting to near 2.0 A ? Any explanation ?
- 2) It is very unusual to have ramachandran outliers at this resolution and 3) the gap between Rfree and Rfactor is large (0.7), suggesting over-refinement. Considering the very good quality of the data collected, the model refinement could be significantly improved so that the geometry is better

(backbone and side chains). Alternatively, an clear explanation with density figure should be provided to explain why these outliers exist.

The crystal structure of the complex solved at a resolution of 3.4 Å has similar issues :

- 1) High MeanI/sigI (4 at the highest resolution shell), which suggests that diffraction could have been exploited at better resolution to a 1.5I/sigI . Again, an explanation is required or better, a more thorough investigation of the data.
- 2) Very large number of outliers = 2.4%, despite R-free being 0.23 (i.e. very low at a 3.4Å resolution). Again, this number is too high and indicates putative over refinement. The model should be improved so that a limited or no ramachandran outliers are present.

P15. The wording in SAXS interpretation is misleading at several instances. Careful rewording is required.

« SAXS analysis of SN9-SH3 alone indicated that the protein is found as a monomer in solution with a structured SH3, forming a common b-sheet structure ». This sentence is wrong. SAXS resolution cannot tell that the two proteins have a common b-sheet structure. In the material and methods it is indicated that alpha fold was used to model the complex. Thus SAXS data is here use to 1) validate the Alfa Fold model and 2) provide the flexible extension with CORAL. This should be clearly indicated in the text. Also provide information on the modelling of the complex with Alpha fold and 2) statistics on its validity (pLDDT). Otherwise, it leaves the impression that the structure was determined experimentally which is not the case. The model was validated and improved with SAXS but is still a model and not a structure. That does not mean that it cannot be interpreted but the wording is important.

Again here a fit of the SAXS curve with the alpha fold model and the coral modified model would show how the latter improves the fit to the SAXS data and why it is necessary.

P16-The sentences a « SAXS analysis of the SemDAPH-SN9-SH3 complex revealed a stoichiometry of 1 :1 with the SH3 domain of SN9 interacting with the PRD1...Sem DPRD1 » should be also modified as well by something like ” the alphafold model predicts that XXX “

P21- in the discussion similar over-interpretation occur : « Our structural analysis revealed that in the « . The structural analysis here suggest but not reveal since there is no high resolution data presented on this complex but instead a AF model and low resolution SAXS. So in absence of experimentally determined structure, this is over-interpretation. Please note that I see no problem with interpreting this model, especially with SAXS data accompanying it but one has to be careful.

Minor : introduce rhodamine-labelled SemD in the text and/or the figure 3 legend so that Fig2b panel with SemD can be understood better.

Reviewer #2 (Remarks to the Author):

The manuscript by Fabienne Kocher and co-workers describes the structural and functional analysis of a protein from the bacterial pathogen *Chlamydia pneumoniae* (Cpn), SemD, which is an early endocytic plasma membrane associated effector protein. SemD, that was previously named CPn0677, recruits G-actin and binds and activates the actin-modulator N-WASP through its C-terminal region for the initiation of branched actin networks via the Arp2/3 complex. These membrane-bound processes enable the developing endocytic vesicle to engulf the infectious elementary body of Cpn, while the associated actin network generates the forces required to reshape and detach the nascent vesicle from the PM.

The authors use X-ray crystallography to determine the structure of the two central WH2-domains and the C-terminal helical tail domain at 2.4 Ang resolution and model the N-terminal proline-rich domains into the assembly based on SAXS analysis. They next analysed the effector complex with N-WASP and succeed in determining a 3.4 Ang structure between SemD and the basic region/GTPase-binding domain (BR-GBD) of N-WASP and identify that SemD interacts with N-WASP in a similar manner as Cdc42-GTP, mimicking thus the small Rho-family GTPase in its active state. The authors use GUV model membranes enriched with PS or PI(4,5)P2 to analyse by confocal light microscopy the association of the N-WASP BR-GBD to vesicles in the presence or absence of SemD, identifying that the basic region is required for SemD-driven membrane binding. As a fourth factor, the Bar-domain containing protein SNX9 comes into play, showing that the N-terminal proline-rich region of SemD interacts with its SH3 domain. A tripartite complex formation of SemD, N-WASP and SNX9 appears feasible, using SEC analysis with recombinant proteins of the interacting domains, sustained by the association to GUVs. The manuscript concludes with a model how the effector protein SemD manipulates the host endocytic machinery to enable it to engulf the large *Chlamydia* elementary body.

The manuscript follows a recent publication from the Hegemann and Molleken groups on the analysis of plasma membrane shaping and co-factor recruitment by SemD (Spona et al., 2023). Overall, although the findings how Cpn SemD hijacks the cytoskeletal machinery of N-WASP, Arp2/3, actin, and links it to the endocytic machinery by SNX9 BAR domain binding and plasma membrane association to facilitate uptake of the Cpn elementary body appear interesting, I wonder if the study will generate enough general interest to justify publication in *Nature Communications*. In addition, although the experimental results appear sound, there is room for improvement in the text and figures, both in the presentation of the data and in the description of the results.

Criticism:

While the authors speak of “structural and functional mimicry” (title) and mention similarities to the Cdc42–N-WASP interaction several times, the only molecular comparison is shown in Supplementary Figure 2b. I would strongly encourage the authors to put this into the main manuscript (somewhere between Fig. 2 and Fig. 3), as the structural comparison explains the mode of action seen in Fig. 2 and the functional depiction the membrane recruitment analysed in Fig. 3.

Figure 3 is not well described, neither in the Figure display nor in the legend. The presence or absence of SemD in the experimental series should be indicated clearly. Which SemD construct

was used? Why is the second row of panel b (BR-GBD in the absence of SemD) not reported in panel c? The figure title seems already interpretation as there is no Cdc42 used in this experiment. I would find something like “SemD recruits the BR-GBD region of N-WASP to membrane vesicles” (phenocopying the Cdc42 activation mechanism) clearer. And the display of the constructs in panel A is strange. The deleted areas (delta181-197 and deltadelta 181-197 / 192?!-213) should be indicated by brackets or lines. At present, the immediate eye-catcher of the BR-GBDdelta construct from panel A would be that it starts at residue 155, but contains mutations in the BR.

I do not appreciate the figure legend description to Fig. 4e, which blurs fact with fiction! “Details of the interaction between the two domains.” It should be added: “Proposed” details ... “, based on the model shown in d.” Although, there is little reason to doubt the interaction diagram, we should still stick to experimentally determined data.

Figure 1b: I have a difficulty with the display of the protein chain of SemB. It almost looks as if there is a knot in the assembly of alpha1 and 2 with alpha 7, 8 and 9. But I think, there is none? How will the G-actin binding WH2 domains (alpha1 and alpha2) unwrap from this fold? Is it correct to name the remaining seven a-helices “a rigid core” (line 120), is it rigid when a1 and a2 unfold? Did the authors perform a DALI search for either a1-a9 or a3-a9 to see if there are similar folds?

Along these lines: What is the role of the two WH2 domains in SemD? Do they interact with G-actin? As far as I can see, it has not been assigned a particular function in the cartoon model of Figure 6. Is the stoichiometry of the SemD–G-actin interaction known? When looking at the 9-helical fold of SemD, maybe the second WH2 domain is not a bona-fide G-actin interactor.

Minor:

The authors speak only of Cdc42 when they mean Cdc42-GTP, as the small GTPase-switch for the interaction with N-WASP.

Figure 1c: Where is the C-terminus of the chain, what points the arrow E382 to, and what sequence follows 382?

A depiction of the N- and C-terminus of a protein chain would be helpful to the not-so-familiar readers of the manuscript. E.g. in Fig. S2b, S3k, 1c

Line 56-58, sentence: “Downstream ..., which have been shown to bind the SH3-domain of SNX9.” A citation is missing here. Who has shown this? It is obviously not a new finding of this study.

Reviewer #3 (Remarks to the Author):

In the current manuscript, Kocher et al., elegantly use crystallography and small-angle x-ray scattering (SAXS) to explore the structural relationship of SemD, a *Chlamydia pneumoniae* (Cpn)

type III effector, and the ability to bind to eukaryotic proteins N-WASP and sorting nexin 9 (SNX9). A previous study, published by the same research team (PMID 37179401), demonstrated that SemD (Cpn0677) is translocated to assist entry of the obligate intracellular pathogen Cpn. Upon translocation, SemD binds to phosphatidylserine that resides in the inner leaflet of the plasma membrane. From this position, SemD recruits members of the endocytic pathway, such as N-WASP and SNX9. Further, they demonstrated that SemD activates N-WASP, which initiates branching actin polymerization. This specific ability of SemD is similar to the activity of active, GTP-bound Cdc42 that can bind to N-WASP to activate actin polymerization. The goal of this current study is to determine the molecular structure of SemD in complex with eukaryotic proteins N-WASP or/and SNX9 to test the hypothesis that SemD can molecularly mimic activated Cdc42. Co-crystallization of SemD and N-WASP reveals that SemD binds to N-WASP in the same regions as activated Cdc42. To experimentally confirm these findings, they used giant unilamellar vesicles (GUVs) containing phosphatidylserine. These types of studies confirmed that SemD binds to N-WASP within the same domains as GTP-Cdc42, and that SemD can simultaneously bind membrane (GUVs), N-WASP, and SNX9. In general, the studies were elegant and rigorous, yet offer an incremental increase to our understanding of the SemD-NWASP-SNX9 complex.

One of the main conclusions by the authors is that SemD can functionally mimic Cdc42, thus making GTP-Cdc42 “superfluous”. In this statement lies the opportunity for the authors to substantially contribute to our knowledge of type III effectors that “mimic” eukaryotic proteins. It is currently unknown if Cpn infection decreases membrane localization/activation of Cdc42. If SemD is a functional mimic of GTP-Cdc42, can SemD outcompete GTP-Cdc42 to bind to N-WASP? If Cdc42 were depleted, could transfected SemD functionally complement Cdc42 activity? Given this research team’s expertise and abilities, answering these scientific inquiries would meaningfully improve the manuscript.

Below are minor concerns that need clarification:

- The sentence in lines 40-42 needs a reference (possibly reference #36 associated with a similar sentence in the Discussion, lines 323-327).
- Pertaining specifically to lines 67-70, but also throughout the manuscript: the description of the various domains of N-WASP is somewhat confusing. Particularly, when referring to the C-terminal part of the GBD domain that is found in the N-terminal segment. Or in lines 163-164 in reference to the CRIB domain. It took several readings to make sure I was understanding that the reference was to N-WASP’s CRIB domain.
- In order to bind to N-WASP, Cdc42 must be active or GTP-bound. Does SemD interact with N-WASP without post-translational modification? This should be clarified or perhaps specific conclusions from the structure can be discussed more thoroughly.
- Please make sure that all abbreviations are defined (e.g. “SEC”, line 150).
- Lines 193-198. Please clarify the differences and similarities of Wiskott-Aldrich syndrome protein (WASP) and neural WASP (N-WASP). It is also unclear why these two distinct proteins are being discussed in this way. Cdc42 and N-WASP have been studied extensively. Further, it is unclear why in Figure S2b, the amino acid sequences being compared between WASP and N-WASP are human versus rat, respectively.
- In line 379, the sentence likely should read: “SemD also binds the BAR-containing protein SNX9,.....”.

- The authors need to explain/justify why a *Rattus norvegicus* clone of N-WASP was used in this study. The previous study used human cells (PMID 37179401).

Replies to reviewers' comments.

Reviewer #1

Remarks to the author:

- 1) *This study provides a very detailed biochemical and structural analysis of the SemD effector function and interaction with host cell partners. The manuscript is well written and the quality of the figures is excellent. The manuscript provide the crystal structure of SemD alone and in complex with N-WASP as well as some SAXS data on three different samples. The GUV binding data, together with mutational analysis, elegantly demonstrates the relevance of the biochemical information gained. I believe this study presents a very significant amount of novel information with some excellent biochemistry. Altogether, it provides detailed mechanistic insights into a single effector tailed to interact multiple cell partners for reprogramming the target cell.*

Generally the interpretation of the data is good and the conclusion supported by the experimental results. However, the authors often refer to these models as a “SAXS structures” while SAXS was mainly used to model unstructured regions and to confirm in silico prediction. Indeed, the authors have used alpha fold but do not mention this in the result section of the manuscript and this is not appropriate and might even be misleading. Otherwise, the methodology is sound except for the crystallography data that requires significant improvement/modification or at least convincing explanations.

We thank the reviewer for these positive comments.

The reservations expressed with regard to the SAXS terminology, the use of AlphaFold, and the methodology applied to the crystallography data are addressed below.

Major:

- 2) *P7: The sentence « Based on the information derived from SAXS, we added the missing N-terminal residues (aa 67-152) to complete the SemDΔAPH structure and the resulting model indeed confirmed that these residues make up the IDR. To draw this conclusion, the authors have used CORAL to model the N-terminal moiety. However, the argument presented as such is circular given that the program CORAL models only flexible region. Similarly, FigS1c as such is not informative because CORAL is made to fit best the SAXS curve so it seems a circular argument. Finally, CORAL can give more than one solution to fit the SAXS, often enabling to illustrate the flexibility. Fig S1c could be modified by showing the fit of the crystal structure AND the CORAL complete model and Fig 1 shows the different possible orientations of the N-terminal. This will allow the reader to understand how the modelling of the N-terminal part is required to fit the SAXS curve, hence providing a more realistic model of the protein. Similar issue occurs for modelling the complex of SemD/SNX9 (see below).*

We thank the reviewer for this comment. We have rephrased this sentence on P7 of the revised MS as follows: “Based on the information derived from SAXS data, we added the missing N-terminal residues (aa 67-137) to complete the SemDΔAPH structure (Fig. 1c). The resulting models showed that these N-terminal residues comprise the IDR tail (the best-fit model is shown in Fig. 1c, an overlay of independent models, showing the same tendency of the tail orientation, are shown in Supplementary Fig. 1h).” (line 135-139). As suggested, we have updated Supplementary Fig. 1 and included a comparison of the theoretical scattering intensity of the solved SemDΔAPH crystal structure in order to illustrate the mismatch relative to the experimental scattering data (Fig S1c, d). Furthermore, we have added an overlay of 10 independent CORAL models in this figure (Figure S1h) to illustrate the possible orientations of the N-terminal tail (χ^2 range: 1.197 – 2.098). We updated the text in the manuscript accordingly (line 138-139) and clarified that Figure 1c shows the best-fit model.

3) *The crystal structure of SemDΔAPH has been solved at 2.4 resolution.*

a) *First it is not clear why the resolutions has been cut at 2.4Å while the I/sI is 5, R_{pim} is 0.15. From these numbers, it seems that the crystal was diffracting to near 2.0 Å? Any explanation?*

Thank you very much for this comment. We agree that we may have been too conservative in our SemDΔAPH data analysis. The dataset contains no symmetry (P1) and within the density the large N-terminal segment is not visible owing to its flexibility as stated on line 116. Therefore, we were very conservative with respect to the resolution cutoff of the data in the original MS. In light of the reviewer's positive comment, we rechecked the initial data, and we now agree with the suggestion that the data extend to 2.1 Å. We have refined the structure of SemDΔAPH and the PDB database kindly allowed us to update our PDB entry (8S5R). Upon comparing the initial structure with the newly refined structure no significant changes were observed, which is also reflected in the low RMSD of 0.3 Å when overlaying. This means that the initial conclusion and interpretation of the structure are still valid.

b) *It is very unusual to have ramachandran outliers at this resolution and 3) the gap between R_{free} and R_{factor} is large (0.7), suggesting over-refinement. Considering the very good quality of the data collected, the model refinement could be significantly improved so that the geometry is better (backbone and side chains). Alternatively, a clear explanation with density figure should be provided to explain why these outliers exist.*

As stated above we have included data up to a resolution of 2.1 Å and carefully refined the structure. We now have better statistics, and the Ramachandran outliers are no longer present. We thank the reviewer for these fruitful comments, which encouraged us to get the best out of the data.

The difference between the R-factor and R-free is still relatively high (albeit with 0.6 slightly lower than initially reported in the original MS). However, this is not unusual and several other structures have been reported with such a difference in the two values. We believe that this is due to the fact that a relatively large N-terminal portion of the protein is flexible and disordered in the crystal.

4) *The crystal structure of the complex solved at a resolution of 3.4 Å has similar issues.*

1) *High MeanI/sigI (4 at the highest resolution shell), which suggests that diffraction could have been exploited at better resolution to a 1.5I/sigI. Again, an explanation is required or better, a more thorough investigation of the data.*

We have closely reevaluated the data and included data at 3.3 Å. Above that resolution, the $R_{\text{merge}} / R_{\text{pim}}$ increases and the I/σ drops simultaneously. This is due to the distance of the detector, which was put at 3.4 Å. We refined the structure also at 3.0 Å but we saw that in this case the refinement statistics deteriorated.

Below we include a log file from XDS from the data statistics. Here the decrease in data quality above 3.3 Å is clear (as indicated by a R_{merge} above 100%). We also think that at this resolution the quality of the maps is unlikely to change significantly between 3.3 Å and 3.2 Å. Therefore, we would like to keep the resolution limit at 3.3 Å.

SUBSET OF INTENSITY DATA WITH SIGNAL/NOISE \geq -3.0 AS FUNCTION OF RESOLUTION

RESOLUTION	NUMBER OF REFLECTIONS	COMPLETENESS	R-FACTOR	R-FACTOR COMPARED	I/SIGMA	R-meas	CC(1/2)	Anomal	SigAno	Nano
LIMIT	OBSERVED	UNIQUE	POSSIBLE	OF DATA	observed	expected	Corr			
13.37	2573	242	242	100.0%	4.4%	5.0%	2573	41.19		
4.6%	99.9*	21	0.836	93						
9.46	5033	443	443	100.0%	4.3%	5.1%	5033	42.87		
4.5%	99.9*	14	0.873	194						
7.72	6435	558	560	99.6%	5.5%	5.9%	6435	35.21		
5.7%	99.9*	8	0.895	248						
6.69	7833	675	677	99.7%	8.2%	8.0%	7833	26.05		
8.6%	99.8*	6	0.935	308						
5.98	8719	748	748	100.0%	12.8%	11.8%	8719	17.97		
13.4%	99.5*	4	0.942	346						
5.46	9556	822	822	100.0%	14.7%	13.9%	9556	15.75		
15.4%	99.4*	5	0.941	381						
5.05	7988	907	920	98.6%	14.5%	13.3%	7987	13.91		
15.4%	99.1*	-8	0.840	427						
4.73	9922	996	996	100.0%	14.7%	13.5%	9921	14.41		
15.5%	99.3*	6	0.926	470						
4.46	10831	1033	1033	100.0%	15.0%	14.1%	10831	14.11		
15.8%	99.3*	3	0.934	485						
4.23	12099	1122	1122	100.0%	18.5%	17.2%	12099	12.53		
19.5%	98.8*	-2	0.904	531						
4.03	12268	1121	1123	99.8%	20.9%	20.1%	12268	10.94		
21.9%	98.9*	7	0.934	532						
3.86	13592	1227	1227	100.0%	26.1%	24.9%	13592	9.02		
27.3%	98.1*	3	0.889	586						
3.71	14321	1280	1282	99.8%	33.3%	33.2%	14321	6.86		
34.9%	97.5*	5	0.859	609						
3.57	14883	1322	1322	100.0%	42.3%	41.3%	14883	5.72		
44.3%	95.7*	5	0.877	629						
3.45	15182	1341	1341	100.0%	58.9%	63.8%	15182	3.90		
61.6%	93.2*	6	0.817	641						
3.34	15884	1397	1399	99.9%	74.5%	80.4%	15884	2.99		
78.0%	88.7*	3	0.791	669						
3.24	16374	1437	1437	100.0%	108.5%	118.8%	16374	1.95		
113.6%	79.3*	-3	0.735	689						
3.15	17242	1517	1517	100.0%	135.2%	156.7%	17242	1.40		
141.6%	64.2*	-4	0.701	732						
3.07	17652	1547	1547	100.0%	176.4%	214.2%	17652	0.90		
184.6%	54.8*	-2	0.635	740						
2.99	18324	1603	1603	100.0%	215.3%	254.3%	18324	0.60		
225.4%	41.4*	-5	0.581	776						
total	236711	21338	21361	99.9%	15.4%	15.9%	236709	9.59		
16.1%	99.9*	2	0.818	10086						

- 2) *Very large number of outliers = 2.4%, despite R-free being 0.23 (i.e. very low at a 3.4Å resolution). Again, this number is too high and indicates putative over refinement. The model should be improved so that a limited or no Ramachandran outliers are present.*

We also re-refined this structure and now the Ramachandran shows one outlier (0.3%) which is due to a crystal contact as stated in the manuscript (see at Materials and Methods “Structure determination via crystallization” on line 809-812). Also, R-free and R-factor are closer together (see Supplementary Table 1). The R-free is still low at this resolution. We believe this is due to the higher resolution structure of SemD, which was used for the phasing and thereby is well defined.

For the reviewer we have included below a picture of the electron density of the interaction site between SemD and BR-GBD. On the left, the overall structure of SemD (colored in blue) and the BR-GBD protein (in green) is presented. In the middle and right panel, we highlighted the density of the interaction site between the two proteins. To indicate the quality of the map we have included now the side chains of the BR-GBD protein. We hope that the reviewer can appreciate that the side chains are clearly visible and that their density is of good enough quality to fit these into the electron density. The electron density of the SemDΔAPH – BR-GBD interface is now also added as Supplemental Figure 8 in the revised version.

Due to the lower resolution of the SemD – BR-GBD complex, we wanted to ensure that the binding, which is based on our SAXS analysis, reveals the same binding mode for the both SemD protein alone and in the complex.

- 5) *P15. The wording in SAXS interpretation is misleading at several instances. Careful rewording is required.*

We carefully reformulated all SAXS-based interpretations to avoid any misunderstandings throughout the manuscript. (e.g. lines 345-347, 495)

- 6) *«SAXS analysis of SN9-SH3 alone indicated that the protein is found as a monomer in solution with a structured SH3, forming a common β -sheet structure». This sentence is wrong. SAXS resolution cannot tell that the two proteins have a common β -sheet structure.*

We thank the reviewer for this comment. This is correct the common β -sheet itself is not a result of the SAXS data. It is rather a common feature of the SH3 domain. The model of SNX9-SH3 we obtained via modeling using AlphaFold contains this motif. This motif is found to be important for binding. We wanted to emphasize that the SAXS modeling also revealed that this β -sheet motif is part of the binding interface.

We have reformulated the sentence as follows: “SAXS analysis of SNX9-SH3 apo indicated that the protein is found as a monomer in solution with a structured core, and a flexible region indicated via the Kratky plot (Supplementary Fig. 6, Supplementary Table 2).” (line 345-347)

- 7) *In the material and methods it is indicated that alpha fold was used to model the complex. Thus SAXS data is here use to 1) validate the Alfa Fold model and 2) provide the flexible extension with CORAL. This should be clearly indicated in the text. Also provide information on the modelling of the complex with Alpha fold and 2) statistics on its validity (p LDDT). Otherwise, it leaves the impression that the structure was determined experimentally which is not the case. The model was validated and improved with SAXS but is still a model and not a structure. That does not mean that it cannot be interpreted but the wording is important.*

Again here a fit of the SAXS curve with the alpha fold model and the coral modified model would show how the latter improves the fit to the SAXS data and why it is necessary.

We thank the reviewer for this comment. We have updated the manuscript (line 347/348) to clarify the fact that we started with an AlphaFold model. Furthermore, we added the initial AlphaFold model as well as the pLDDT plot in Figure S7a,b and also include a comparison with the theoretical scattering intensity of the AlphaFold model to illustrate the mismatch relative to the experimental scattering data (Figure S7c,d). We have clarified in the text that we improved this starting model with CORAL (line 364-371).

- 8) *P16-The sentences a « SAXS analysis of the SemDAPH-SN9-SH3 complex revealed a stoichiometry of 1 :1 with the SH3 domain of SN9 interacting with the PRD1...Sem DPRD1 » should be also modified as well by something like " the alphafold model predicts that XXX "*

We thank the reviewer for this comment. In lines 362-364 we shortened this sentence to "The SAXS analysis of the SemDAPH – SNX9-SH3 complex confirmed a stoichiometry of 1:1, based on the molecular weight." and added the AlphaFold model interpretation earlier in the manuscript (line 347-362).

- 9) *P21- in the discussion similar over-interpretation occur: « Our structural analysis revealed that in the « . The structural analysis here suggest but not reveal since there is no high resolution data presented on this complex but instead a AF model and low resolution SAXS. So in absence of experimentally determined structure, this is over-interpretation. Please note that I see no problem with interpreting this model, especially with SAXS data accompanying it but one has to be careful.*

We thank the reviewer for this comment. We have recast the sentence in the Discussion as follows: "Our model analysis suggests that, in the flexible N-terminal half of SemD, which is separated by a linker sequence from the rigid core that mediates N-WASP interaction, five residues in the PRD1 domain interact with six residues in the β -sheet structure of the SNX9-SH3 domain (Figs. 5 and 7c)". (lines 495-498)

Minor

- 10) *Introduce rhodamine-labelled SemD in the text and/or the figure 3 legend so that Fig2b panel with SemD can be understood better.*

We checked Figure 3 and the corresponding text passage and introduced SemD^{Rhod} for rhodamine-labelled SemD in the text (line 238) and in the Figure 3 legend (line 573/574).

Reviewer #2

Remarks to the Author:

The manuscript by Fabienne Kocher and co-workers describes the structural and functional analysis of a protein from the bacterial pathogen Chlamydia pneumoniae (Cpn), SemD, which is an early endocytic plasma membrane associated effector protein. SemD, that was previously named CPn0677, recruits G-actin and binds and activates the actin-modulator N-WASP through its C-terminal region for the initiation of branched actin networks via the Arp2/3 complex. These membrane-bound processes enable the developing endocytic vesicle to engulf the infectious elementary body of Cpn, while the associated actin network generates the forces required to reshape and detach the nascent vesicle from the PM.

The authors use X-ray crystallography to determine the structure of the two central WH2-domains and the C-terminal helical tail domain at 2.4 Ang resolution and model the N-terminal

proline-rich domains into the assembly based on SAXS analysis. They next analysed the effector complex with N-WASP and succeed in determining a 3.4 Ang structure between SemD and the basic region/GTPase-binding domain (BR-GBD) of N-WASP and identify that SemD interacts with N-WASP in a similar manner as Cdc42-GTP, mimicking thus the small Rho-family GTPase in its active state. The authors use GUV model membranes enriched with PS or PI(4,5)P2 to analyse by confocal light microscopy the association of the N-WASP BR-GBD to vesicles in the presence or absence of SemD, identifying that the basic region is required for SemD-driven membrane binding. As a fourth factor, the Bar-domain containing protein SNX9 comes into play, showing that the N-terminal proline-rich region of SemD interacts with its SH3 domain. A tripartite complex formation of SemD, N-WASP and SNX9 appears feasible, using SEC analysis with recombinant proteins of the interacting domains, sustained by the association to GUVs. The manuscript concludes with a model how the effector protein SemD manipulates the host endocytic machinery to enable it to engulf the large Chlamydia elementary body.

The manuscript follows a recent publication from the Hegemann and Molleken groups on the analysis of plasma membrane shaping and co-factor recruitment by SemD (Spona et al., 2023). Overall, although the findings how Cpn SemD hijacks the cytoskeletal machinery of N-WASP, Arp2/3, actin, and links it to the endocytic machinery by SNX9 BAR domain binding and plasma membrane association to facilitate uptake of the Cpn elementary body appear interesting, I wonder if the study will generate enough general interest to justify publication in Nature Communications. In addition, although the experimental results appear sound, there is room for improvement in the text and figures, both in the presentation of the data and in the description of the results.

Criticism:

1. While the authors speak of “structural and functional mimicry” (title) and mention similarities to the Cdc42–N-WASP interaction several times, the only molecular comparison is shown in Supplementary Figure 2b. I would strongly encourage the authors to put this into the main manuscript (somewhere between Fig. 2 and Fig. 3), as the structural comparison explains the mode of action seen in Fig. 2 and the functional depiction the membrane recruitment analyzed in Fig. 3.

We thank the reviewer for this suggestion. We have now inserted the Supplementary Fig. 2b into Fig. 3 (see panel 3a).

2. Figure 3 is not well described, neither in the Figure display nor in the legend. The presence or absence of SemD in the experimental series should be indicated clearly.

We really appreciate this comment.

Figure 3 now consists of 4 panels (see point 1 above). Accordingly, the original Fig. 3a-c are now labelled 3b-d. We have made sure that the absence or presence of SemD is clearly indicated in both the relevant figure and the figure legend (see below).

Which SemD construct was used?

In Figure 3, the full-length Rhodamine-labelled SemD (SemD^{Rhod}) was used as now indicated in Figure 3c and d. In the main text, we also introduced SemD^{Rhod} in line 238.

Why is the second row of panel b (BR-GBD in the absence of SemD) not reported in panel c?

For panel c (previously panel b), we now mention in more detail the individual constructs as well as the negative controls used. In panel c we also denoted the presence and absence of SemD in more detail. Moreover, in panel d (previously panel c), the data for BR-GBD_{GFP} in the absence of SemD^{Rhod} have been added. Finally, the individual samples in the panel c and d have been re-arranged in such a way, that the first GUV experiment in panel c (BR-GBD_{GFP} + SemD^{Rhod}) corresponds to the first set of boxplots in panel d, and so on.

The figure title seems already interpretation as there is no Cdc42 used in this experiment. I would find something like “SemD recruits the BR-GBD region of N-WASP to membrane vesicles” (phenocopying the Cdc42 activation mechanism) clearer.

We have modified the title of figure 3 as proposed. (Line 559)

And the display of the constructs in panel A is strange. The deleted areas (delta181-197 and deltadelta 181-197 / 192?!-213) should be indicated by brackets or lines. At present, the immediate eye-catcher of the BR-GBDdelta construct from panel A would be that it starts at residue 155, but contains mutations in the BR.

The original panel a now is panel b, as we have incorporated the structural comparison of the binding of Cdc42_{GTP} and SemD to WASP and N-WASP respectively, as panel a. Thus, BR-GBD and its deletion variants are now represented in panel b. As suggested by the reviewer, we have changed the depictions of the BR-GBD deletion variants by marking the deleted protein regions by dashed lines in square brackets (with the first and last deleted amino acids being indicated). These alterations are also specified in the figure legend on line 571-573.

3. *I do not appreciate the figure legend description to Fig. 4e, which blurs fact with fiction! “Details of the interaction between the two domains.” It should be added: “Proposed” details ... “; based on the model shown in d.” Although, there is little reason to doubt the interaction diagram, we should still stick to experimentally determined data.*

The original Figure 4 is now Figure 5. We agree with the reviewer. We have altered the figure legend to Figure 5e accordingly (line 619-620).

4. *Figure 1b: I have a difficulty with the display of the protein chain of SemB. It almost looks as if there is a knot in the assembly of alpha1 and 2 with alpha 7, 8 and 9. But I think, there is none? How will the G-actin binding WH2 domains (alpha1 and alpha2) unwrap from this fold? Is it correct to name the remaining seven a-helices “a rigid core” (line 120), is it rigid when a1 and a2 unfold?*

We thank the reviewer for pointing this out. The crystal structure of SemD reveals an arrangement of nine intertwined alpha-helices, which together form a rigid core. The reviewer is right that helices 1 and 2 carry the predicted WH2_1 and WH2_2 sequences, respectively. Experimentally, we previously showed that full-length SemD binds G-actin *in vitro*, while its N-terminal (aa 1 – 137) and C-terminal (aa 218 – 382) portions – both of which lack the predicted WH2 domains, do not bind G-actin¹. These findings demonstrate that the central SemD region (aa 138 – 216), which includes the predicted WH2_1 and WH2_2 sequences, is essential for G-actin binding. However, the stoichiometry of this interaction was not clear. The SemD structure now tells us, that the predicted WH2_1 sequence (aa 138 – 178) within alpha helix 1 (aa 155 to aa 177) is largely available for interaction with G-actin, while parts of the WH2_2 sequence (aa 179 – 216) in alpha helix two (aa 179 to aa 203) are not completely available (see Fig. 1b, c). This suggests that

WH2_1 is responsible for G-actin binding. Comparison of the SemD core region alone and in the presence of the BR-GBD fragment of N-WASP reveals an almost identical conformation (RMSD of 0.7 Å, line 172-174), suggesting that SemD serves as a stable platform for its various interactions. Thus, currently we speculate that all nine SemD alpha helices form a rigid core during its interactions with host proteins.

To make the localization of these helices clearer, we have changed the helical depiction in Figure 1b to a cylindrical presentation, which we hope gives a better view of the positions of all helices relative to each other. We have added a statement concerning the possible role of WH_1 in G-actin binding to the discussion (lines 511-517).

Did the authors perform a DALI search for either a1-a9 or a3-a9 to see if there are similar folds?

We searched for similar folds by performing an EBI fold search (which is similar to a DALI search) as well as a DALI search using a1-a9 or a3-a9. However, the best hits consisted of structures in which (i) the RMSD is rather high (starting at 4 Å) and (ii) the overlapping regions were rather small (in total 40-70 amino acids). Thus, SemD appears to possess a unique arrangement of 9 helices not found in other proteins. We have added a sentence to this effect in the revised manuscript (lines 122-124).

5. *Along these lines: What is the role of the two WH2 domains in SemD? Do they interact with G-actin? As far as I can see, it has not been assigned a particular function in the cartoon model of Figure 6. Is the stoichiometry of the SemD–G-actin interaction known? When looking at the 9-helical fold of SemD, maybe the second WH2 domain is not a bona-fide G-actin interactor.*

We thank the reviewer for this suggestion. As already explained above under point 4, we previously published experimental *in vitro* evidence that the central region of SemD carrying the two putative WH2 sequences is essential for G-actin binding¹. The stoichiometry of the SemD – G-actin interaction is not known. As suggested, we have now added G-actin to our model in Figure 7 (previously Fig. 6). As already stated under point 4, the accessibility options available to the two WH2 sequences possibly suggest that WH2_1 might be the G-actin binding sequence. We have added a statement concerning the possible role of WH2_1 in G-actin binding in the discussion (lines 511-517).

Minor:

6. *The authors speak only of Cdc42 when they mean Cdc42-GTP, as the small GTPase-switch for the interaction with N-WASP.*

We thank the reviewer for this comment. Throughout the MS, we have replaced “Cdc42” by “Cdc42_{GTP}” for referring to the active form, where required. (Line 71)

7. *Figure 1c: Where is the C-terminus of the chain, what points the arrow E382 to, and what sequence follows 382?*

We modified the figure legend to explaining the amino acid labelling at the N- and C-termini of the SemD structures shown in Fig. 1b and 1c (lines 533-535 and lines 539-540).

8. A depiction of the N- and C-terminus of a protein chain would be helpful to the not-so-familiar readers of the manuscript. E.g. in Fig. S2b, S3k, 1c

We have added the depictions of the N- and/or C-termini as follows:

Fig. 1c		A clearer Figure description in the figure legend
Fig. 2	Added	b&d: BR-GBD C-terminus b: SemD N-terminus
	Not added	d: SemD C-terminus (is buried)
Fig. 3	Added	a: N- and C-terminus of BR-GBD
Fig. 5	Added	d: N- and C-terminus of SemD
Fig. 6	Added	c: C-terminus of BR-GBD and SNX9-SH3 in the right panel
	Not added	c: C-terminus of SNX9 and SemD in the left panel, C-terminus of BR-GBD and SemD in the right panel (are buried)
Fig. S3k	Added	Model 1: N- and C-termini for BR-GBD and SemD Model 2: C-termini of BR-GBD and SemD, N-terminus of SemD Model 3: N-terminus of SemD and C-terminus of BR-GBD
	Not added	Model 2: N-terminus of BR-GBD (buried) Model 3: C-terminus of SemD and N-terminus of BR-GBD (buried)
Fig. S7a	Added	N-termini of SemD and SNX9 C-terminus of SNX9
	Not added	C-terminus of SemD (buried)

9. Line 56-58, sentence: "Downstream ..., which have been shown to bind the SH3-domain of SNX9." A citation is missing here. Who has shown this? It is obviously not a new finding of this study.

We thank the reviewer for this comment. We reformulated the sentence and have now included the relevant citation (Line 54).

Reviewer #3

Remarks to the Author:

In the current manuscript, Kocher et al., elegantly use crystallography and small-angle x-ray scattering (SAXS) to explore the structural relationship of SemD, a Chlamydia pneumoniae (Cpn) type III effector, and the ability to bind to eukaryotic proteins N-WASP and sorting nexin 9 (SNX9). A previous study, published by the same research team (PMID 37179401), demonstrated that SemD (Cpn0677) is translocated to assist entry of the obligate intracellular pathogen Cpn. Upon translocation, SemD binds to phosphatidylserine that resides in the inner leaflet of the plasma membrane. From this position, SemD recruits members of the endocytic pathway, such as N-WASP and SNX9. Further, they demonstrated that SemD activates N-WASP, which initiates branching actin polymerization. This specific ability of SemD is similar to the activity of active, GTP-bound Cdc42 that can bind to N-WASP to activate actin polymerization. The goal of this current study is to do determine the molecular structure of SemD in complex with eukaryotic proteins N-WASP or/and SNX9 to test the hypothesis that SemD can molecularly mimic activated Cdc42. Co-crystallization of SemD and N-WASP reveals that SemD binds to N-WASP in the same regions as activated Cdc42. To experimentally confirm these findings, they used giant unilamellar vesicles (GUVs) containing phosphatidylserine. These types of studies confirmed that SemD binds to N-WASP within the same domains as GTP-Cdc42, and that SemD can simultaneously bind membrane (GUVs),

N-WASP, and SNX9. In general, the studies were elegant and rigorous, yet offer an incremental increase to our understanding of the SemD-NWASP-SNX9 complex.

Major:

1. *One of the main conclusions by the authors is that SemD can functionally mimic Cdc42, thus making GTP-Cdc42 “superfluous”. In this statement lies the opportunity for the authors to substantially contribute to our knowledge of type III effectors that “mimic” eukaryotic proteins. It is currently unknown if Cpn infection decreases membrane localization/activation of Cdc42. If SemD is a functional mimic of GTP-Cdc42, can SemD outcompete GTP-Cdc42 to bind to N-WASP? If Cdc42 were depleted, could transfected SemD functionally complement Cdc42 activity? Given this research team’s expertise and abilities, answering these scientific inquiries would meaningfully improve the manuscript.*

We thank the reviewer for bringing up this very insightful suggestion. The question of whether SemD structurally and functionally mimics Cdc42_{GTP} in activating N-WASP is also able to activate other Cdc42_{GTP}-controlled target proteins is indeed very interesting. However, the situation is complex. The small GTPase Cdc42 is involved in a wide variety of different cellular processes, such as the cell cycle, gene transcription, regulation of the cytoskeleton, cell movement, and cell polarization, and novel activities are still being found^{2,3}.

Small GTPases such as Cdc42 are subject to complex regulation. *In vitro*, GTP hydrolysis and GDP–GTP exchange of Cdc42 are extremely slow. *In vivo*, these steps are accelerated by GAPs (GTPase activating proteins) and GEFs (guanine nucleotide exchange factors), respectively. The GTPase cycle has a distinct spatial context within the cell, with active (GTP-bound) Cdc42 taking place between the membrane of the endoplasmic reticulum and the plasma membrane via their carboxy-terminal prenyl groups. Following inactivation, the GTPase can be extracted from the plasma membrane by RhoGDI (Rho guanine nucleotide dissociation inhibitor), which encloses the prenyl group and maintains the inactive GTPase in a soluble, cytosolic form⁴. It has been shown that expression of fluorescently tagged Rho GTPase fusion proteins does not properly reflect the normal enzymes localization or function. Furthermore, expression of exogenous Rho GTPases can upset the stoichiometric balance of the Rho GTPases with RhoGDI, resulting in aggregation and degradation of the GTPases (see Box 1 in Bement et al.⁴). Similarly, knockdown of a Rho-GTPase releases RhoGDI1, with consequences for other Rho-GTPases⁵. Thus, manipulation of Rho protein levels will impair the entire Rho network with consequences for the biological activities controlled by Rho proteins.

For these reasons, we find a cellular approach to studying Cdc42 localization during a *Cpn* infection challenging. This also holds for depletion of Cdc42 followed by functional complementation by transfected SemD. Instead, we decided on a biochemical approach to tackle these questions (lines 272-320).

Motivated by the reviewer’s comment,

- (i) We first tested whether SemD can outcompete active Cdc42_{GTP} for binding to N-WASP. In pull-down experiments using recombinant proteins, we incubated BR-GBD-GFP-His with SemD Δ APH and Cdc42, bound to a non-hydrolyzing nucleotide analogue (Cdc42_{GppNHp}). Using GFP-Trap® agarose we showed that SemD Δ APH outcompetes Cdc42_{GppNHp} for binding to BR-GBD. This result is now included in the revised MS in Figure 4a and 4b.
- (ii) To verify this result we then performed stopped-flow experiments in which we first allowed BR-GBD to form a complex with Cdc42_{mGppNHp}, and then added an equimolar amount of SemD Δ APH. We observed an immediate change in relative fluorescence, indicating that the added SemD Δ APH competes with Cdc42_{mGppNHp} for binding to BR-GBD, and eventually leads to

the active dissociation of Cdc42_{mGppNHp} from BR-GBD. These data are included in the revised MS in Figure 4c and 4d.

The results from (i) and (ii) indicate that SemD has an improved binding capacity for N-WASP than does active Cdc42_{GTP}, and can thus outcompete the latter.

- (iii) The regulation of the actin cytoskeleton, e.g. by activating N-WASP and Formins, is a major function for Cdc42. Thus, the autoinhibition of Formins is released upon binding of active Cdc42_{GTP} to the Formin GTPase binding domain⁶. Therefore, we decided to test whether SemD could mimic Cdc42_{GTP} in binding to Formin. We tested for direct interaction of recombinant SemDΔAPH with ForminL2. In a pulldown experiment with GST-agarose, using FMNL2 fused to GST as bait, we first showed that Cdc42_{GppNHp}, which we used as positive control, showed significant binding to FMNL2_{GST}. Remarkably, SemD does not bind to FMNL2_{GST}. These data indicate that SemD specifically binds the Cdc42 effector N-WASP, but not FMNL2. This data set is included in the revised MS in Figure 4a and 4f.

Thus, our new data indicate that the effector protein SemD has evolved in *Cpn* specifically for interaction with N-WASP.

Minor:

Below are minor concerns that need clarification:

2. *The sentence in lines 40-42 needs a reference (possibly reference #36 associated with a similar sentence in the Discussion, lines 323-327).*

We have added the relevant reference. (Line 42)

3. *Pertaining specifically to lines 67-70, but also throughout the manuscript: the description of the various domains of N-WASP is somewhat confusing. Particularly, when referring to the C-terminal part of the GBD domain that is found in the N-terminal segment. Or in lines 163-164 in reference to the CRIB domain. It took several readings to make sure I was understanding that the reference was to N-WASP's CRIB domain.*

We thank the reviewer for this comment. We agree that, owing to the complexity of the N-WASP domains, the description sometimes might be misleading. We have rephrased the description of the various N-WASP domains and now refer to Figure 2a in the introduction, where a schematic representation of the individual domains and their composition can be found (lines 63-67). We hope that the modified description of the N-WASP domains is now easier to understand (e.g. on lines 179-183 and throughout the MS).

4. *In order to bind to N-WASP, Cdc42 must be active or GTP-bound. Does SemD interact with N-WASP without post-translational modification? This should be clarified or perhaps specific conclusions from the structure can be discussed more thoroughly.*

We thank the reviewer for this interesting question. Based on our structural data, we are certain that there is no post-translational modification of SemD occurring during binding to N-WASP.

In the case of nucleotide-binding proteins, for example Cdc42, during recombinant expression in *E. coli*, a corresponding nucleotide is directly attached to the protein. After protein purification, this nucleotide is removed and replaced by the specific nucleotide required (see M&M for details, lines 685-705).

Moreover, we expressed all our SemD constructs in *E. coli* as well; however, we found no evidence for an attached nucleotide or any other post-translational modification either in our crystal structure or by our SAXS studies.

5. *Please make sure that all abbreviations are defined (e.g. "SEC", line 150).*

- SAXS, pIDDT, SEC, IMAC, SDS/PAGE, methods part!

We have modified the MS accordingly and have defined all abbreviations upon their first mentioning (e.g. SEC on line 163-164, SAXS on line 93-94, pLDDT on line 351-352 and throughout the entire MS)

6. *Lines 193-198. Please clarify the differences and similarities of Wiskott-Aldrich syndrome protein (WASP) and neural WASP (N-WASP). It is also unclear why these two distinct proteins are being discussed in this way. Cdc42 and N-WASP have been studied extensively. Further, it is unclear why in Figure S2b, there the amino acid sequences being compared between WASP and N-WASP are human versus rat, respectively.*

We thank the reviewer for this comment and the opportunity to clarify this point. As far as we know the structure of the complex formed between active Cdc42_{GTP} and N-WASP is not known. As yet, only the structure of the interaction between WASP and active Cdc42_{GTP} has been elucidated⁷. Therefore, we compared our SemD – N-WASP BR-GBD structure with the published Cdc42_{GTP} – WASP BR-GBD₂₃₀₋₂₈₈ structure obtained by NMR⁷. We now noted in the text that a structure for the Cdc42_{GTP} – N-WASP complex is not yet available (lines 213-214).

In the MS, we state that WASP and N-WASP belong to the same protein family, share 56% identity and 74% similarity, and exhibit strikingly similar domain architectures and regulatory mechanisms, including release of autoinhibition by binding of active Cdc42_{GTP} (lines 211-217). To make the point clearer, we also moved the Supplemental Figure 2b, which shows a sequence comparison of the relevant BR-GBD domains from WASP_{human} and N-WASP_{rat}, from the Supplemental information to Figure 3a.

Finally, we performed all experiments with N-WASP from *Rattus norvegicus*, as this protein was also used in our previous work about SemD¹. Importantly, the SemD and Cdc42_{GTP} interacting domains of N-WASP_{human} and N-WASP_{rat} are identical (see sequence comparison below). The entire BR-GBD domain in N-WASP_{human} and N-WASP_{rat} carries 3 aa changes, all of which lie outside of the interface between SemD and human N-WASP BR-GBD domain in a region that is not resolved in our SemD – N-WASP BR-GBD structure (see below).

7. *In line 379, the sentence likely should read: “SemD also binds the BAR-containing protein SNX9,……”.*

We adapted the sentence accordingly (now line 493).

8. *The authors need to explain/justify why a Rattus norvegicus clone of N-WASP was used in this study. The previous study used human cells (PMID 37179401).*

In the previous SemD study¹, all plasmid constructs used for protein expression in *E. coli* or for transfection in human cells carried the N-WASP cDNA from *Rattus Norvegicus* obtained from Addgene.

Importantly, and as already stated under point 6 (see above): the amino-acid sequences of the regions of N-WASP BR-GBD (dark grey box) that interact with Cdc42_{GTP} and SemD respectively are identical in N-WASP_{rat} and N-WASP_{human} (see sequence alignment below). The N-WASP BR-GBD segments found in the two species differ in only three positions (marked by two red boxes) in the region N-terminal to the BR segment; the structure of this N-terminal region was not resolved in our co-crystal. Thus, the conclusions from this MS hold true for human N-WASP as well.

We have added the following statement to the M+M section: The N-WASP BR-GBD protein fragment from *R. norvegicus* and *human* differ by 3 amino acids located N-terminal to the BR domain outside of our co-crystal structure (lines 672-674)

Sequence alignment of N-WASP_{human} and N-WASP_{rat}

Sequence alignment of N-WASP_{human} and N-WASP_{rat}. | The BR-GBD fragment used in this study for complex formation with SemDΔAPH is marked in pink and the individual domains (BR, CRIB and C-sub) are marked with lines on top of the corresponding sequence. The binding region of BR-GBD, as resolved by the crystal structure, is marked with a grey box and the BR-GBD fragment resolved by crystallography is marked with a black arrow. Inside the binding region, N-WASP_{human} and N-WASP_{rat} are identical, whereas the whole fragment (labelled pink) used for this study shows 3 amino acid exchanges (red boxes), all being N-terminally to the BR-motif.

References

- 1 Spona, D., Hanisch, P. T., Hegemann, J. H. & Molleken, K. A single chlamydial protein reshapes the plasma membrane and serves as recruiting platform for central endocytic effector proteins. *Commun Biol* **6**, 520 (2023).
- 2 Bustelo, X. R., Sauzeau, V. & Berenjeno, I. M. GTP-binding proteins of the Rho/Rac family: regulation, effectors and functions in vivo. *Bioessays* **29**, 356-370 (2007).
- 3 Fu, J. *et al.* The role of cell division control protein 42 in tumor and non-tumor diseases: A systematic review. *J Cancer* **13**, 800-814 (2022).
- 4 Bement, W. M., Goryachev, A. B., Miller, A. L. & von Dassow, G. Patterning of the cell cortex by Rho GTPases. *Nat Rev Mol Cell Biol* **25**, 290-308 (2024).
- 5 Boulter, E. *et al.* Regulation of Rho GTPase crosstalk, degradation and activity by RhoGDI1. *Nat Cell Biol* **12**, 477-483 (2010).
- 6 Peng, J., Wallar, B. J., Flanders, A., Swiatek, P. J. & Alberts, A. S. Disruption of the Diaphanous-related formin Drf1 gene encoding mDia1 reveals a role for Drf3 as an effector for Cdc42. *Curr Biol* **13**, 534-545 (2003).
- 7 Abdul-Manan, N. *et al.* Structure of Cdc42 in complex with the GTPase-binding domain of the 'Wiskott-Aldrich syndrome' protein. *Nature* **399**, 379-383 (1999).

REVIEWERS' COMMENTS

Reviewer #1 (Remarks to the Author):

The authors have considerably improved the X-ray crystallography data processing, also clarified the contributions of modelling, structural and SAXS in their statements and have addressed all my questions.

I have no further comment on this excellent manuscript.

Reviewer #2 (Remarks to the Author):

This is a well-executed revision of the study on the *Chlamydia pneumoniae* SemD protein binding to the actin-modulator N-WASP. I have no further comments and suggest publication of this manuscript.

Reviewer #3 (Remarks to the Author):

The authors were highly responsive to reviewer comments. I have no more concerns about the manuscript.